# MicroRNA-26b protects against MASH development in mice and can be efficiently targeted with lipid nanoparticles

Linsey Peters[1,2,3,4†], Leonida Rakateli[1,2†], Rosanna Huchzermeier[1,2], Andrea Bonnin-Marquez[1,2], Sanne L Maas[1,2], Cheng Lin[5,6], Alexander Jans[5], Yana Geng[7], Alan Gorter[7], Marion Gijbels[3,8], Sander Rensen[9], Peter Olinga[7], Tim Hendrikx[10], Marcin Krawczyk[11], Malvina Brisbois[12], Joachim Jankowski[1,2,3], Kiril Bidzhekov[4], Christian Weber[4,13,14,15,16], Erik AL Biessen[1,3], Ronit Shiri-Sverdlov[17], Tom Houben[17], Yvonne Döring[4,13,18], Matthias Bartneck[5,19,20], Emiel van der Vorst[1,2,4,21]*

[1]Institute for Molecular Cardiovascular Research (IMCAR), RWTH Aachen University, Aachen, Germany; [2]Aachen-Maastricht Institute for CardioRenal Disease (AMICARE), RWTH Aachen University, Aachen, Germany; [3]Department of Pathology, Cardiovascular Research Institute Maastricht (CARIM), Maastricht University, Maastricht, Netherlands; [4]Institute for Cardiovascular Prevention (IPEK), Ludwig-Maximilians-Universität München, Munich, Germany; [5]Department of Rheumatology and Shanghai Institute of Rheumatology, Renji, Shanghai, China; [6]Department of Medicine III, University Hospital Aachen, Aachen, Germany; [7]Department of Pharmaceutical Technology and Biopharmacy, Groningen Research Institute of Pharmacy, University of Groningen, Groningen, Netherlands; [8]Department of Medical Biochemistry, Amsterdam Cardiovascular Sciences: Atherosclerosis & Ischemic Syndrome; Amsterdam Infection and Immunity: Inflammatory diseases; Amsterdam UMC location University of Amsterdam, Amsterdam, Netherlands; [9]Department of Surgery, NUTRIM, School of Nutrition and Translational Research in Metabolism, Maastricht University, Maastricht, Netherlands; [10]Department of Laboratory Medicine, Medical University Vienna, Vienna, Austria; [11]Department of Gastroenterology, Hepatology and Transplant Medicine, Medical Faculty, University of Duisburg, Essen, Germany; [12]Department of Medicine II, Saarland University Medical Center, Saarland University, Homburg, Germany; [13]DZHK (German Center for Cardiovascular Research), Partner Site Munich Heart Alliance, Munich, Germany; [14]Department of Biochemistry, Cardiovascular Research Institute Maastricht (CARIM), Maastricht University, Maastricht, Netherlands; [15]Munich Cluster for Systems Neurology (SyNergy), Munich, Germany; [16]Cluster for Nucleic Acid Therapeutics Munich (CNATM), Munich, Germany; [17]Department of Genetics and Cell Biology, School of Nutrition and Translational Research in Metabolism (NUTRIM), University of Maastricht, Maastricht, Netherlands; [18]Swiss Cardiovascular Center, Division of Angiology, Inselspital, Bern University Hospital, University of Bern, Bern, Switzerland; [19]DWI – Leibniz Institute for Interactive Materials, Aachen, Germany; [20]Institute of Technical and Macromolecular Chemistry, RWTH Aachen University, Aachen, Germany; [21]Department of Internal Medicine I - Cardiology, University Hospital, RWTH Aachen University, Aachen, Germany

*For correspondence:
evandervorst@ukaachen.de

†These authors contributed equally to this work

Competing interest: The authors declare that no competing interests exist.

## eLife Assessment

This study presents **valuable** insights into the involvement of miR-26b in the progression of metabolic dysfunction-associated steatohepatitis (MASH). The delivery of microRNA-containing nanoparticles to reduce MASH severity has practical implications as a therapeutic strategy. The authors use two sets of transgenic mouse models, conducted kinase activity profiling of mouse liver samples, and supplemented their findings with additional experiments on human liver and plasma, providing **solid** support for their findings.

**Abstract** The prevalence of metabolic dysfunction-associated steatohepatitis (MASH) is increasing, urging more research into the underlying mechanisms. MicroRNA-26 (*Mir26b*) might play a role in several MASH-related pathways. Therefore, we aimed to determine the role of *Mir26b* in MASH and its therapeutic potential using *Mir26b* mimic-loaded lipid nanoparticles (LNPs). *Apoe*$^{-/-}$*Mir26b*$^{-/-}$, *Apoe*$^{-/-}$*Lyz2*$^{cre}$*Mir26b*$^{fl/fl}$ mice, and respective controls were fed a Western-type diet to induce MASH. Plasma and liver samples were characterized regarding lipid metabolism, hepatic inflammation, and fibrosis. Additionally, *Mir26b* mimic-loaded LNPs were injected in *Apoe*$^{-/-}$*Mir26b*$^{-/-}$ mice to rescue the phenotype and key results were validated in human precision-cut liver slices. Finally, kinase profiling was used to elucidate underlying mechanisms. *Apoe*$^{-/-}$*Mir26b*$^{-/-}$ mice showed increased hepatic lipid levels, coinciding with increased expression of scavenger receptor a and platelet glycoprotein 4. Similar effects were found in mice lacking myeloid-specific *Mir26b*. Additionally, hepatic TNF and IL-6 levels and amount of infiltrated macrophages were increased in *Apoe*$^{-/-}$*Mir26b*$^{-/-}$ mice. Moreover, *Tgfb* expression was increased by the *Mir26b* deficiency, leading to more hepatic fibrosis. A murine treatment model with *Mir26b* mimic-loaded LNPs reduced hepatic lipids, rescuing the observed phenotype. Kinase profiling identified increased inflammatory signaling upon *Mir26b* deficiency, which was rescued by LNP treatment. Finally, *Mir26b* mimic-loaded LNPs also reduced inflammation in human precision-cut liver slices. Overall, our study demonstrates that the detrimental effects of *Mir26b* deficiency in MASH can be rescued by LNP treatment. This novel discovery leads to more insight into MASH development, opening doors to potential new treatment options using LNP technology.

## Introduction

'Metabolic dysfunction-associated steatotic liver disease' (MASLD), formerly known as non-alcohol fatty liver disease (NAFLD; *Eslam et al., 2020*), is the most common form of fatty liver disease. It accounts for roughly 25% of liver disease cases globally and is defined by fat accumulation in the liver in the absence of heavy alcohol consumption (*Eslam et al., 2020*). MASLD refers to a group of disorders that include simple metabolic dysfunction-associated steatotic liver (MASL) and metabolic dysfunction-associated steatohepatitis (MASH; *Eslam et al., 2020*). An estimated 20–25% of MASL patients will acquire MASH, and both MASL and MASH prevalence are predicted to rise even higher over the next decade due to increased frequency of risk factors such as obesity and insulin resistance (*Peng et al., 2020*). While MASL is still considered reversible and generally benign, its progression into MASH is thought to be harmful in most cases since it precedes the development of liver fibrosis, cirrhosis, and cancer (*Peng et al., 2020*). Moreover, the underlying mechanisms that initiate inflammation and cause MASL to progress into MASH are still poorly understood, restricting the range of available treatments.

One of the main players that may be involved in these processes is microRNAs. MicroRNAs are short non-coding RNAs that regulate gene expression post-transcriptionally by limiting mRNA translation (*Gjorgjieva et al., 2019*). Because microRNAs control numerous genes and processes, multiple microRNAs have been demonstrated to play a role in fundamental aspects of MASH, including lipid metabolism, hepatic inflammation, and fibrosis (*Gjorgjieva et al., 2019*; *López-Sánchez et al., 2021*).

Although many microRNAs have been examined in MASLD, the possible pathogenic involvement of numerous others, remains unknown. One of the microRNAs that has been researched in a number of pathologies, including cancer, cardiovascular illnesses, and neurological disorders, is *Mir26b* (*van der Vorst et al., 2021*). However, very little is known about the function of *Mir26b* in

**eLife digest** Fatty liver disease is a condition characterized by the abnormal accumulation of fat in the liver. In certain cases, the fatty build-up can lead to inflammation and, in time, scarring. This advanced stage is known as MASH (short for metabolic dysfunction-associated steatohepatitis), and it can increase the risk of liver failure, cancer, and other complications. Yet the underlying mechanisms that initiate inflammation and thereby drive the disease are still poorly understood. Identifying the molecular factors contributing to this transition could aid in discovering new treatment targets.

To explore this question, Peters et al. focus on microRNA-26b, a small molecule involved in many heart and metabolic diseases that helps regulate gene expression. They aimed to clarify the role of microRNA-26b in MASH using mice genetically manipulated to lack this regulatory molecule. The experiments revealed that the animals had larger amounts of fat in their livers, with the organs also showing clear signs of scarring and increased inflammation – including high levels of inflammatory signalling molecules and the presence of immune cells known as macrophages.

Peters et al. then treated the animals with specially designed compounds that can act as microRNA-26b. The molecules were safely delivered to the liver within tiny fat-based spheres known as lipid nanoparticles. Following such treatment, the mice showed decreased levels of liver fat and inflammation. The anti-inflammatory effect of the microRNA-26b 'mimics' was also confirmed in human liver samples.

Together, these results show that microRNA-26b plays a protective role in the development of MASH. Future research should focus on confirming whether these molecules could represent a viable therapeutic treatment, in particular when delivered within lipid-based nanoparticles.

the liver and MASH. Nonetheless, some studies have already revealed that *Mir26b* may play a role in several MASH-related pathways. For example, *Mir26b* appears to suppress the nuclear factor-kappa B (NF-κB) pathway in human hepatocellular carcinoma cell lines by inhibiting the expression of TGF-activated kinase-1 (TAK1) and TAK1-binding protein-3, both of which are positive regulators of the NF-κB system (*Zhao et al., 2014*). Furthermore, *Mir26b* has been shown to downregulate the gene expression of platelet-derived growth factor receptor-beta (PDGFR-β), hence reducing hepatic fibrogenesis (*Yang et al., 2019*).

The findings above show that *Mir26b* may play a role in the pathogenesis of MASH, although no study has determined its exact causal involvement in MASH development. In this investigation, we used previously described *Mir26b* knockout mice (*van der Vorst et al., 2021*) and myeloid cell-specific *Mir26b* knockout mice to clarify the role of *Mir26b* in hepatic lipid metabolism, inflammation, and fibrogenesis. Furthermore, lipid nanoparticles (LNPs), which act as clinically and therapeutically relevant vehicles (*Jeong et al., 2023*), loaded with *Mir26b* mimics were used to restore *Mir26b* levels in mice as well as in human precision-cut liver slices to investigate its therapeutic potential.

## Results

### Mice deficient in *Mir26b* show increased hepatic lipid levels and an increased expression of hepatic lipid uptake receptors

To determine the role of *Mir26b* in hepatic lipid metabolism and MASH development, hepatic cholesterol and triglyceride levels were measured in $Apoe^{-/-}$ and $Apoe^{-/-}Mir26b^{-/-}$ mice that were fed a 4-week WTD (*Figure 1A*). Total cholesterol and triglyceride levels were significantly increased in the livers of $Apoe^{-/-}Mir26b^{-/-}$ mice compared to control mice (*Figure 1B–C*). This hepatic lipid effect was also confirmed by Oil-red-O staining, showing increased lipid accumulation in the livers of $Apoe^{-/-}Mir26b^{-/-}$ mice (*Figure 1D–E*). Moreover, pathological scoring of the Oil-red-O staining unveiled that the $Apoe^{-/-}Mir26b^{-/-}$ mice showed a clear tendency towards increased steatosis compared to controls, especially of macrovesicular steatosis (*Figure 1F*).

To elucidate possible mechanisms underlying the increased hepatic lipid levels, gene expression levels of key proteins involved in lipid metabolism were measured in liver tissues of $Apoe^{-/-}$ and $Apoe^{-/-}Mir26b^{-/-}$ mice. Notably, a knockout of *Mir26b* did not affect the expression levels of 'ATP binding cassette subfamily A member 1' (*Abca1*) or 'acetyl-CoA acetyltransferase 2' (*Acat2*; *Figure 1G–H*).

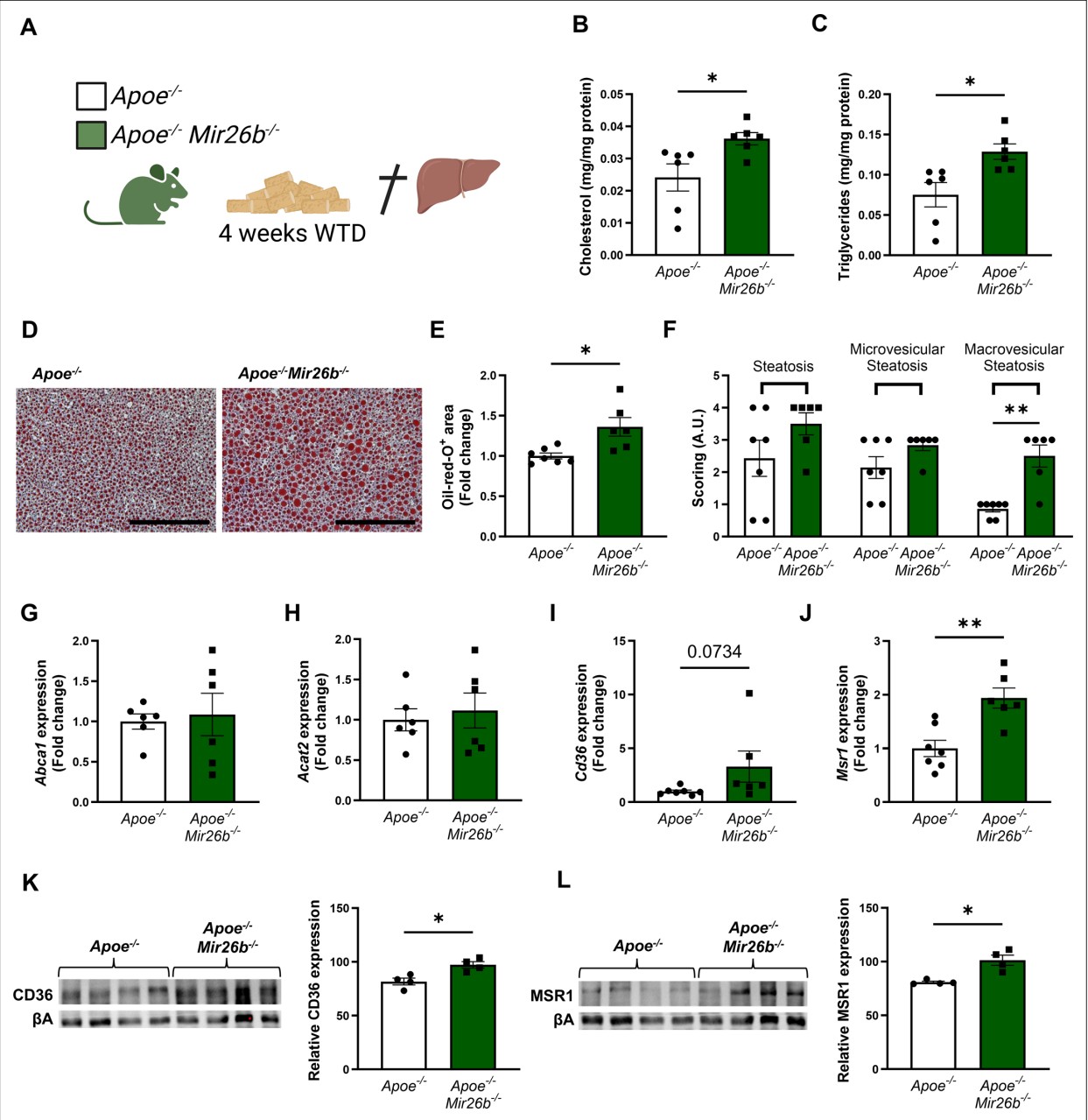

**Figure 1.** Hepatic lipid levels and the expression of lipid uptake receptors are increased by a whole-body knockout of *Mir26b* in mice. (**A**) Schematic overview of the experimental approach. This panel was created using BioRender.com. (**B–C**) Hepatic total cholesterol (**B**) and triglyceride (**C**) measurements normalized against total protein. (**D**) Representative pictures of Oil-red-O staining of liver sections. Scale bar = 200 µm. (**E**) Quantification of the Oil-red-O staining. (**F**) Pathological scoring of the Oil-red-O staining. (**G–J**) Gene expression analysis of (**G**) *Abca1*, (**H**) *Acat2*, (**I**) *Cd36*, and (**J**) *Msr1*. (**K–L**) Western-blot analysis and quantification of (**K**) CD36 and (**L**) MSR1. Fold change is corrected for sex. *p<0.05; **p<0.01. n=4–7 animals per group.

The online version of this article includes the following source data for figure 1:

**Source data 1.** Raw data from *Figure 1*.

However, livers of *Apoe⁻/⁻Mir26b⁻/⁻* mice showed a clearly increased expression of scavenger receptor *Cd36* (*Figure 1I*) and a striking twofold increase of the expression of macrophage scavenger receptor 1 (*Msr1*; *Figure 1J*). This could also be validated on protein level, by showing increased CD36 as well as MSR1 expression in livers of *Apoe⁻/⁻Mir26b⁻/⁻* mice (*Figure 1K–L*). Since these scavenger receptors are highly expressed on macrophages, we have evaluated the contribution of myeloid *Mir26b* to the

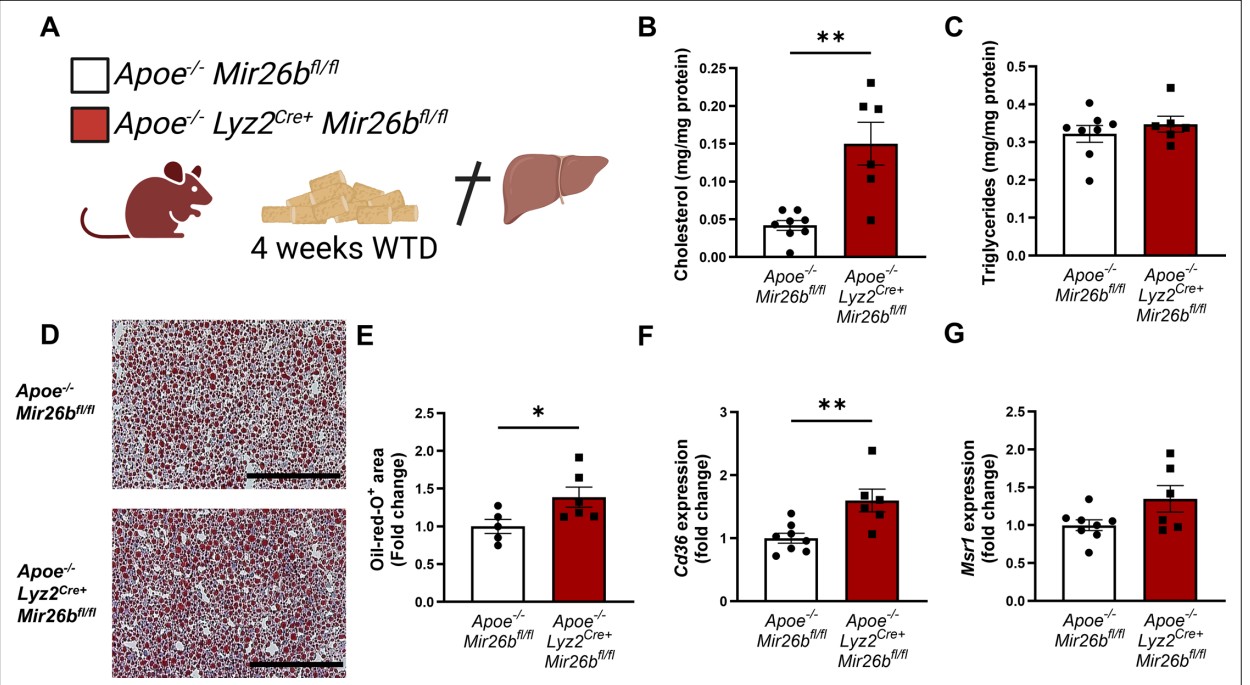

**Figure 2.** Hepatic lipid levels and the expression of lipid uptake receptors are increased by a myeloid-specific *Mir26b* deficiency in mice. (**A**) Schematic overview of the experimental approach. This panel was created using BioRender.com. (**B–C**) Hepatic total cholesterol (**B**) and triglyceride (**C**) measurements normalized against total protein. (**D**) Representative pictures of Oil-red-O staining of liver sections. Scale bar = 200 μm. (**E**) Quantification of the Oil-red-O staining. (**F–G**) Gene expression analysis of (**F**) *Cd36* and (**G**) *Msr1*. Fold change is corrected for sex. *p<0.05; **p<0.01. n=6–8 animals per group.

The online version of this article includes the following source data for figure 2:

**Source data 1.** Raw data from *Figure 2*.

observed hepatic lipid effects. Interestingly, mice that have a myeloid-specific lack of *Mir26b* also show increased hepatic cholesterol levels and lipid accumulation demonstrated by Oil-red-O staining, coinciding with an increased hepatic *Cd36* expression (*Figure 2*), demonstrating that myeloid *Mir26b* plays a major, but not exclusive, role in the observed steatosis.

## Livers of *Mir26b* knockout mice have higher levels of inflammatory cytokines and an increased number of infiltrating macrophages

Besides lipid accumulation, an increased inflammatory profile is a key characteristic of MASH (*Peng et al., 2020*). Therefore, we aimed to elucidate the role of *Mir26b* in hepatic inflammation. Hepatic protein levels of the pro-inflammatory cytokines IL-6 and TNF were significantly increased in *Apoe*[-/-] *Mir26b*[-/-] mice compared to controls (*Figure 3A–B*), while levels of the chemokines CCL2 and CXCL1 remained unchanged (*Figure 3C–D*). To further investigate the effects of the whole-body knockout on a cellular level, liver sections were stained to identify several key leukocyte subpopulations.

Mac-1-positive cells were significantly increased in livers of mice lacking *Mir26b*, indicating a higher infiltration of macrophages and neutrophils (*Figure 3E*). To identify whether the increase of Mac-1-positive cells is due to macrophage or neutrophil infiltration we also determined the number of Ly6G-positive cells, which remained unchanged, suggesting that the increased number of Mac-1-positive cells was likely the result of an increase in the number of infiltrating macrophages rather than neutrophils (*Figure 3F*). Furthermore, the whole-body knockout of *Mir26b* only affected the number of infiltrating macrophages and not Kupffer cells, which are recognized as CD68-positive cells (*Figure 3G*). Furthermore, the number of CD3-positive cells did not differ between *Apoe*[-/-]*Mir26b*[-/-] mice and controls (*Figure 3H*), suggesting that *Mir26b* does not affect T-cell counts in the liver.

Collectively, these results indicate that *Mir26b* plays a protective role in hepatic inflammation by influencing TNF and IL-6 levels and macrophage infiltration in the liver.

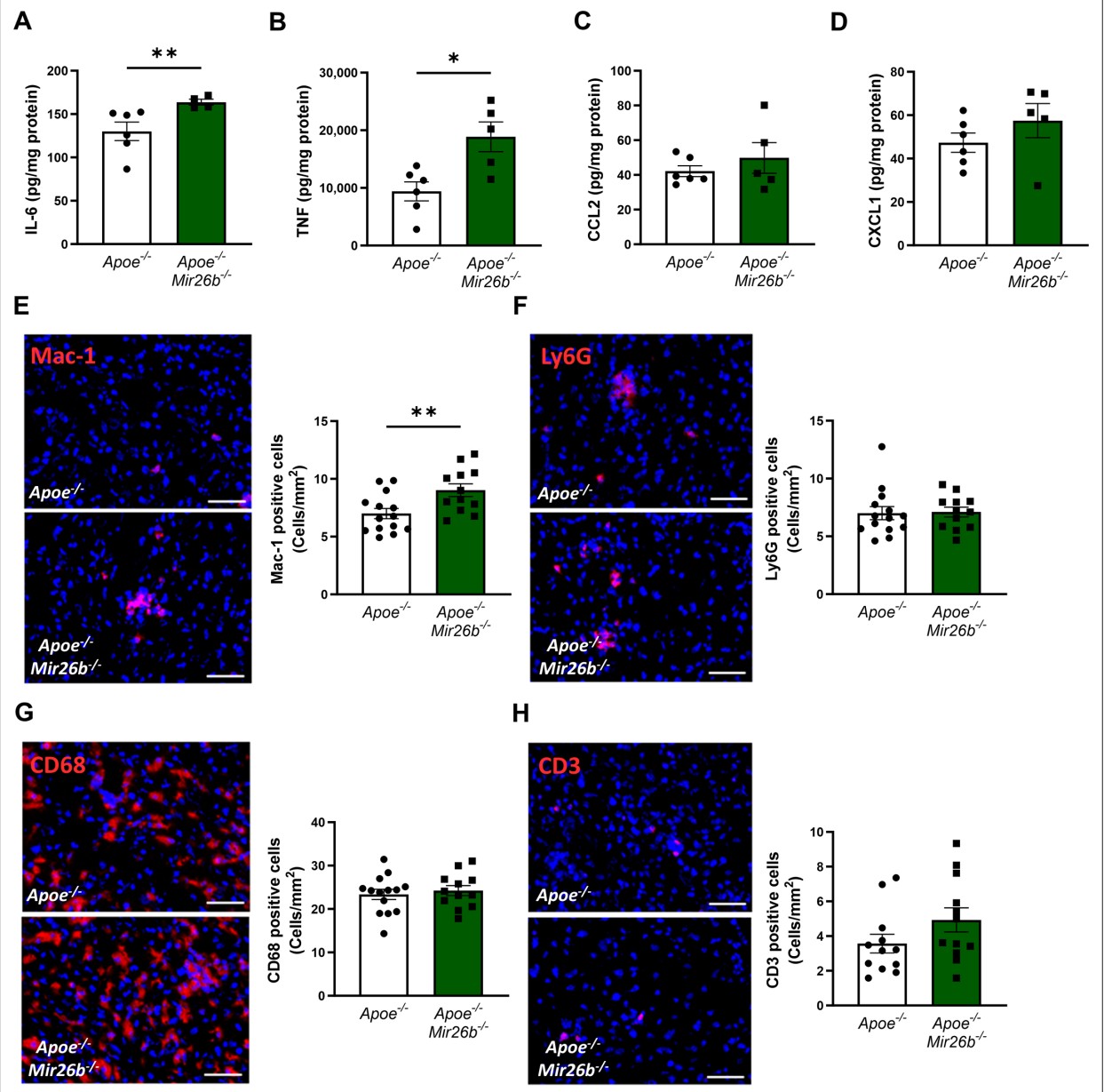

**Figure 3.** Livers of *Apoe*[-/-] *Mir26b*[-/-] mice show elevated pro-inflammatory cytokine levels and an increased number of Mac-1-positive cells. (**A–D**) Cytokine levels of (**A**) IL-6, (**B**) TNF, (**C**) CCL2, and (**D**) CXCL1 were measured in liver protein lysates. (**E–H**) Representative images and quantification of immunofluorescent stainings for (**E**) infiltrating macrophages and neutrophils (Mac-1), (**F**) neutrophils (Ly6G), (**G**) resident monocytes/macrophages (CD68), and (**H**) T-cells (CD3). Scale bar = 50 μm. *p<0.05; **p<0.01. n=6–7 animals per group.

The online version of this article includes the following source data for figure 3:

**Source data 1.** Raw data from *Figure 3*.

## A knockout of *Mir26b* in mice results in increased hepatic fibrosis, which coincides with an increased expression of *Tgfb*

Continued hepatic inflammation can cause fibrotic changes in the liver, which is another characteristic of MASH (*Peng et al., 2020*). As such, we also investigated the influence of *Mir26b* on hepatic fibrosis in mice. Collagen deposition in liver sections was determined by a Sirius-red staining, which showed that a knockout of *Mir26b* significantly exacerbated hepatic fibrosis (*Figure 4A–B*). This was further supported by the increased expression of 'transforming growth factor β' (*Tgfb*) in the livers of *Apoe*[-/-] *Mir26b*[-/-] mice compared to controls (*Figure 4C*). Another gene involved in liver fibrosis, that is 'actin

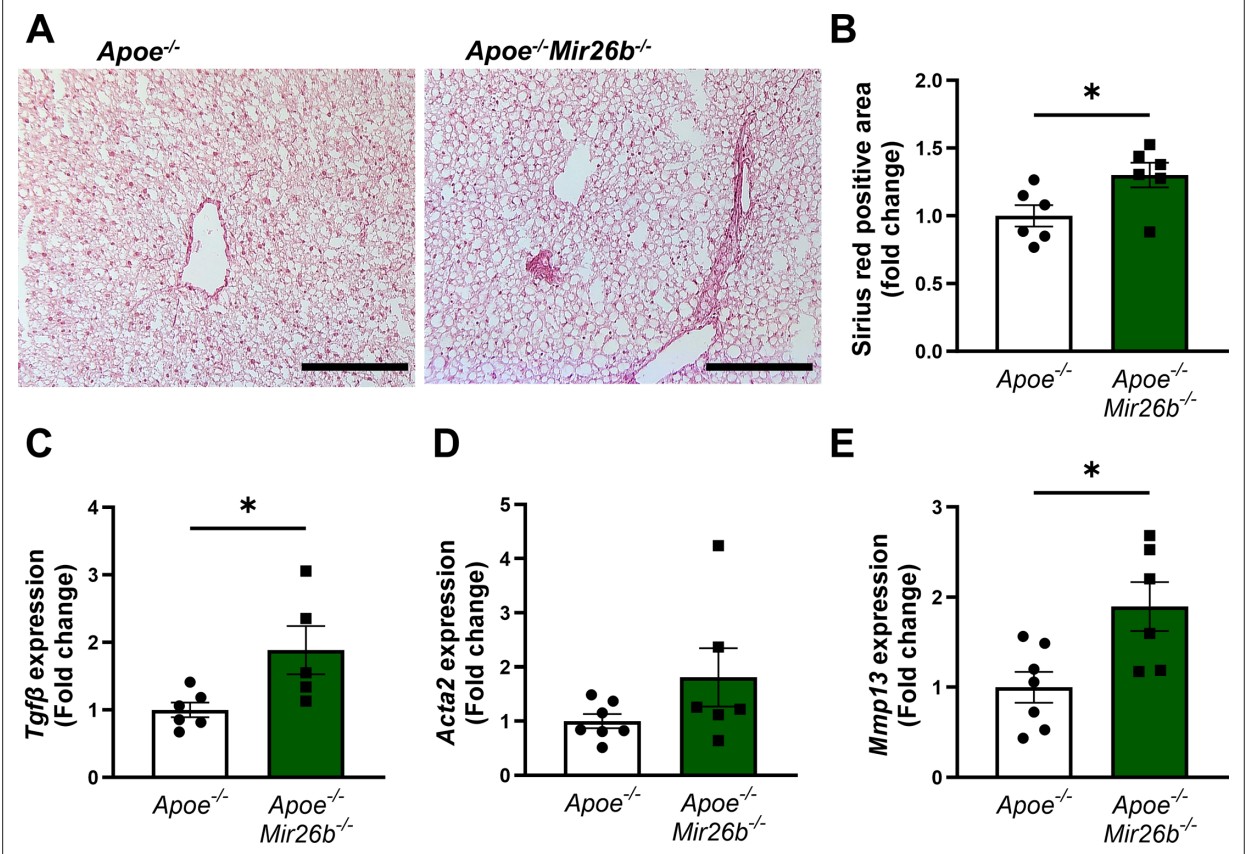

**Figure 4.** Livers of *Apoe⁻/⁻Mir26b⁻/⁻* mice show increased hepatic fibrosis. (**A**) Representative pictures of Sirius-red staining of liver sections. Scale bar = 100 μm. (**B**) Quantification of the Sirius-red staining. (**C–E**) Gene expression of (**C**) *Tgfb*, (**D**) *Acta2*, and (**E**) *Mmp13*. Fold change is corrected for sex. *p<0.05. n=6–7 animals per group.

The online version of this article includes the following source data for figure 4:

**Source data 1.** Raw data from *Figure 4*.

alpha 2′ (*Acta2*), trended towards an elevated expression in mice lacking *Mir26b* (*Figure 4D*). Lastly, a whole-body knockout of *Mir26b* resulted in an increased expression of 'matrix metalloproteinase 13' (*Mmp13*; *Figure 4E*). Overall, these results imply a protective role of *Mir26b* in liver fibrosis, which is linked to an altered expression of *Tgfb*.

## Liver of *Mir26b* knockout mice show highly increased kinase activity related to inflammatory pathways

To elucidate the underlying mechanisms behind the effects of *Mir26b* on the liver, we have performed a kinase activity profiling, focusing on serine-threonine kinases (STK). In order to evaluate the differentially activated kinases, the degree of phosphorylation of peptides coated on STK arrays is determined. Liver lysates from *Apoe⁻/⁻Mir26b⁻/⁻* mice (KO) showed a strong and very distinct upregulation of peptide phosphorylation compared to liver lysates from *Apoe⁻/⁻* mice (WT; *Figure 5A–B*). Remarkably, 84 kinases were significantly more activated in liver lysates from *Apoe⁻/⁻Mir26b⁻/⁻* mice compared to controls (*Figure 5C*), many of which are involved in inflammatory pathways such as c-Jun-N-terminal kinases (JNKs), mitogen-activated protein kinases (MAPKs), and extracellular-signal, regulated kinases (ERKs). This was also further corroborated by pathway analysis (*Figure 5D–E*), showing enrichment in pathways related to inflammation (e.g. MAP kinase activation, TLR signaling) and angiogenesis (e.g. VEGF signaling). Combined, these results demonstrate that the lack of *Mir26b* increases kinase activity in the liver, particularly of kinases related to inflammatory pathways, which can thus be a plausible mechanism behind the hepatic effects observed in *Mir26b*-deficient mice.

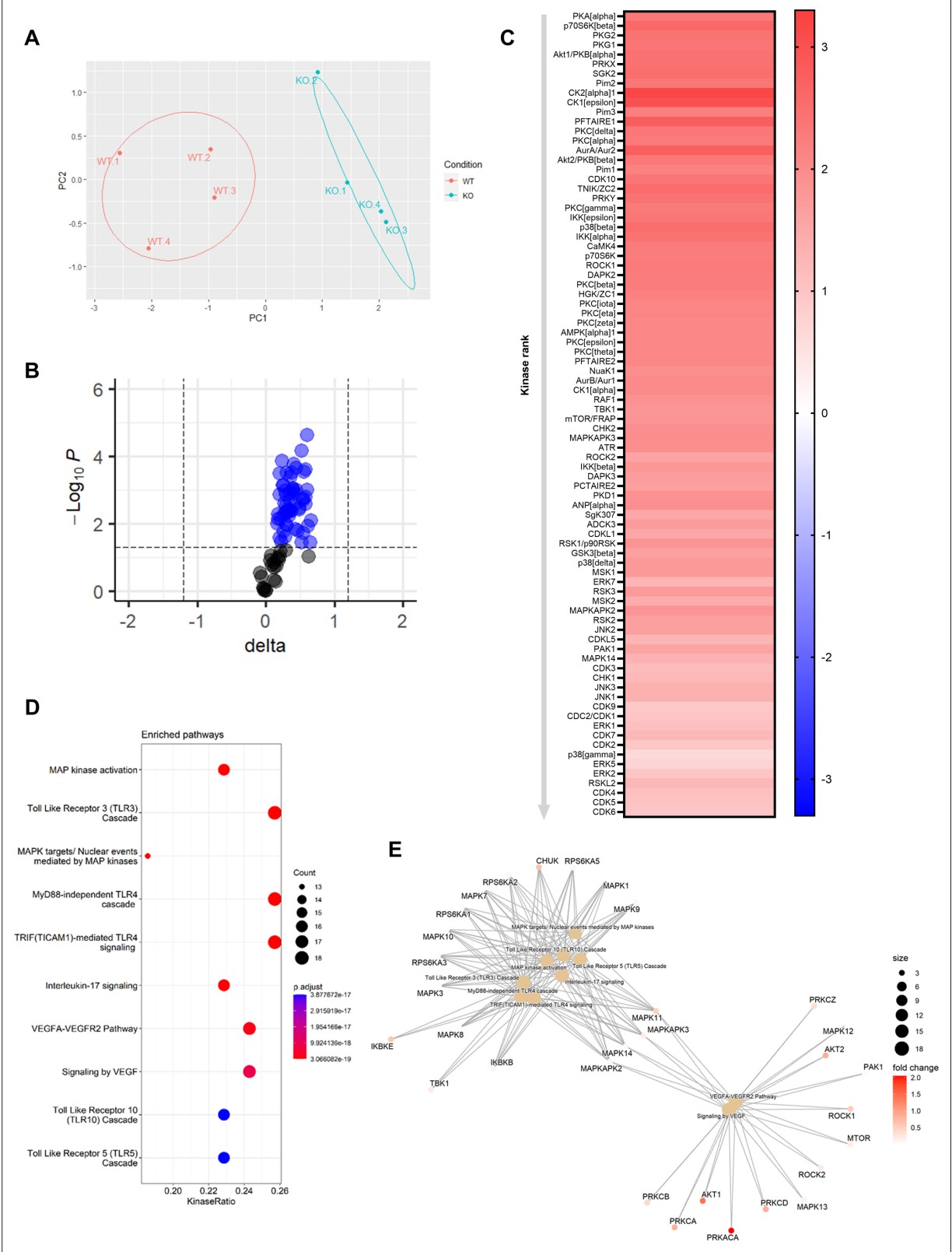

**Figure 5.** Knockout of *Mir26b* results in an increased hepatic inflammatory kinase activity. (**A**) Principal component analysis (PCA) of phosphorylated peptides from STK array (n=4) of liver lysates from *Apoe⁻/⁻Mir26b⁻/⁻* mice (KO) or *Apoe⁻/⁻* (WT) mice. (**B**) Volcano plot visualizing fold change and p-value for phosphorylated peptides from STK array. Blue dots represent significantly altered phosphopeptides. (**C**) Heatmap of significantly changed kinases are ranked based on Median Final Score (cut-off value of 1.2), STK array performed on liver lysates from *Apoe⁻/⁻Mir26b⁻/⁻* mice (KO) compared to *Apoe⁻/⁻*

*Figure 5 continued on next page*

*Figure 5 continued*

(WT) mice. Color corresponds to the Median Kinase Statistic, which represents effect size and directionality (red = increased activity in KO vs. WT mice). (**D**) Enriched pathways based on STK array. (**E**) Network diagram of the pathway enrichment analysis.

The online version of this article includes the following source data for figure 5:

**Source data 1.** Raw data from *Figure 5*.

## Lipid nanoparticles loaded with *Mir26b* mimics can partly rescue the MASH phenotype in whole-body *Mir26b* knockout mice

Since the whole-body knockout mouse model demonstrated that *Mir26b* plays a role in MASH, we attempted to rescue the phenotype by injecting *Apoe*$^{-/-}$*Mir26b*$^{-/-}$ mice on WTD with LNPs, which act as clinically and therapeutically relevant vehicles (*Jeong et al., 2023*), loaded with *Mir26b* mimics (mLNPs) and empty lipid nanoparticles (eLNPs) as control for 4 weeks (*Figure 6A*). These mLNPs over-express the *Mir26b* level in the whole-body deficient mouse (*Figure 6—figure supplement 1A–B*), providing insight into the therapeutic potential of *Mir26b*. Injections with mLNPs lowered hepatic cholesterol levels compared to the vehicle control (*Figure 6B*), whilst triglyceride levels remained unaffected (*Figure 6C*). These findings were further confirmed by demonstrating that treatment with mLNPs significantly reduced the Oil-red-O positive area (*Figure 6D–E*). While mLNP treatment did not affect *Cd36* expression (*Figure 6F*), it resulted in a 0.67-fold reduction in *Msr1* expression compared to mice injected with eLNPs (*Figure 6G*). Although mLNP treatment affects hepatic lipid levels, no changes could be observed on hepatic inflammation, characterized by inflammatory cyto-kines and infiltrated myeloid cells, or fibrosis (*Figure 6H–L*). However, expression of *Tgfb* was signifi-cantly reduced upon mLNP treatment (*Figure 6M*), coinciding with a tendency to decreased *Mmp13* expression (*Figure 6N*). These changes on gene expression suggest that the mLNP treatment might have been to short to observe pronounced effects on later stages of disease development like inflam-mation and fibrosis.

Furthermore, kinase activity profiling of liver lysates demonstrated a distinct downregulation of peptide phosphorylation upon mLNP treatment of *Apoe*$^{-/-}$*Mir26b*$^{-/-}$ (KO.LNP) mice (*Figure 7A–B* and *Figure 7—figure supplement 1A–B*). Interestingly, principal component analysis (PCA) clearly demonstrated that livers from mLNP-treated *Apoe*$^{-/-}$*Mir26b*$^{-/-}$ (KO.LNP) mice more closely resem-bled *Apoe*$^{-/-}$ (WT) mice rather than *Apoe*$^{-/-}$*Mir26b*$^{-/-}$ (KO) mice (*Figure 7A*). The kinase activity of 76 kinases in the liver was significantly downregulated upon mLNP treatment of *Apoe*$^{-/-}$*Mir26b*$^{-/-}$ mice (*Figure 7—figure supplement 1C*). The notion that 60 (79%) of these downregulated kinases were originally upregulated by the *Mir26b* deficiency (*Figure 7C*), furthermore indicates that mLNP treat-ment rescues the observed effects due to the *Mir26b* deficiency. In line with this, pathway analysis also showed an enrichment of similar pathways as described before, that is pathways related to inflamma-tion and angiogenesis (*Figure 7D–E*), again demonstrating that there are at least already early signs that inflammation is influenced by mLNP treatment.

Overall, treatment with mLNPs attenuated MASH development with regard to hepatic lipids and inflammatory kinase activity, highlighting the therapeutic potential of LNPs loaded with *Mir26b* mimics.

## Lipid nanoparticles loaded with *Mir26b* mimics have anti-inflammatory effects on human livers

Since the mouse experiments demonstrated a clear therapeutic potential of LNPs loaded with *Mir26b* mimics, we also set out to explore this potential in a human setting. Human precision-cut liver slices were cultured for 24 hr or 48 hr in the presence of mLNPs or eLNPs as control (*Figure 8A*). Although no effects were observed on IL-6 secretion (*Figure 8B*), mLNPs had strong anti-inflammatory effects on these human precision-cut liver slices. The secretion of TNF, CCL2, and CXCL1 was significantly reduced in slices treated with mLNPs compared to eLNP-treated liver slices (*Figure 8C–E*), under-lining the clear potential of these *Mir26b*-loaded LNPs in a human context.

To further evaluate the importance of *Mir26b* in liver diseases in humans, we have measured the expression levels of *Mir26b*-3p and *Mir26b*-5p in the plasma of healthy subjects and patients with liver cirrhosis (*Figure 8F*). Remarkably, both *Mir26b*-3p and –5 p were significantly elevated in the plasma of liver cirrhosis patients (*Figure 8G–H*), suggesting – at least – a strong association between *Mir26b* and the development of MASH in humans.

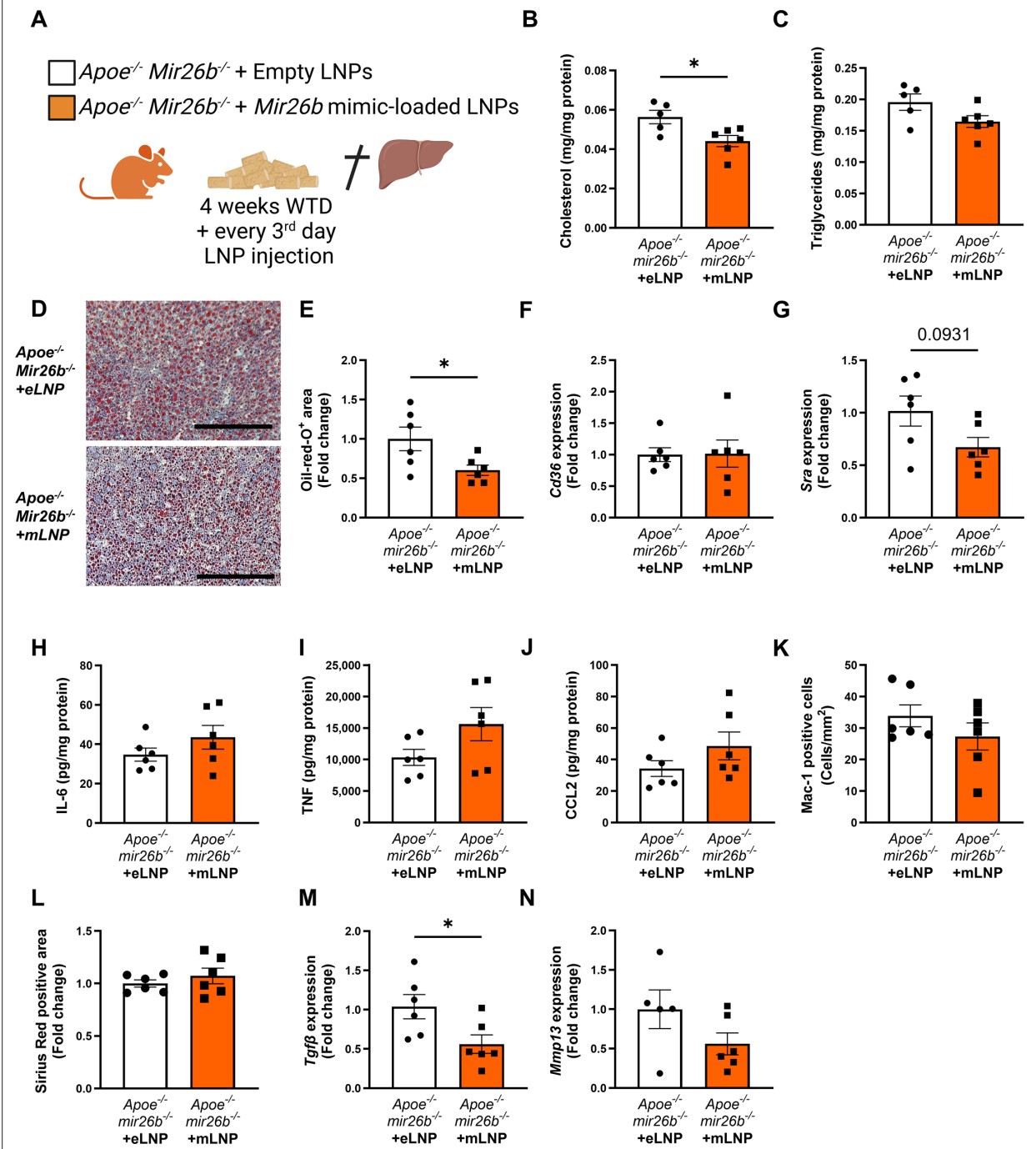

**Figure 6.** *Apoe⁻/⁻Mir26b⁻/⁻* mice injected with LNPs containing *Mir26b* mimics show decreased hepatic lipid levels compared to vehicle control injected mice. (**A**) Schematic overview of the experimental approach in which mice on 4 week WTD were simultaneously injected every 3 days with either empty LNPs as vehicle control (eLNP) or LNPs containing *Mir26b* mimics (mLNP). This panel was created using BioRender.com. (**B–C**) Hepatic total cholesterol (**B**) and triglyceride (**C**) measurements normalized against total protein. (**D**) Representative pictures of Oil-red-O staining of liver sections. Scale bar = 200 µm. (**E**) Quantification of the Oil-red-O staining. (**F–G**) Gene expression analysis of (**F**) *Cd36* and (**G**) *Msr1*. (**H–J**) Cytokine levels of (**H**) IL-6, (**I**) TNF, and (**J**) CCL2 were measured in liver protein lysates. (**K**) Quantification of immunofluorescent staining for infiltrating macrophages and neutrophils (Mac-1). (**L**) Quantification of the Sirius-red staining. (**M–N**) Gene expression of (**M**) *Tgfb*, and (**N**) *Mmp13*. Fold change is corrected for sex. *p<0.05. n=6 animals per group.

The online version of this article includes the following source data and figure supplement(s) for figure 6:

**Source data 1.** Raw data from *Figure 6*.

*Figure 6 continued on next page*

*Figure 6 continued*

**Figure supplement 1.** mLNP treatment overexpresses *Mir26b*-3p and –5 p in murine livers.

**Figure supplement 1—source data 1.** Source data contains the raw data from *Figure 6—figure supplement 1*.

## Discussion

MicroRNAs have been indicated to play a critical role in the development of several pathologies, including MASH. While the role of *Mir26b* has already been investigated in various cardio-metabolic diseases (*van der Vorst et al., 2021*), its role in MASH remained unknown so far. Therefore, this study focused on the role of *Mir26b* in MASH showing that mice deficient in *Mir26b* presented with a higher hepatic lipid content, higher levels of pro-inflammatory cytokines, and an increased number of infiltrated macrophages in the liver and exacerbated hepatic fibrosis. This coincided with an increased activity of kinases that are involved in inflammatory pathways. Finally, when attempting to rescue this phenotype by injecting LNPs loaded with *Mir26b* mimics, we managed to decrease hepatic cholesterol, overall hepatic lipid levels, gene expression of *Msr1*, and the activity of inflammatory kinases. These anti-inflammatory effects of *Mir26b* loaded LNPs were also confirmed in human precision-cut liver slices. Taken together, these results suggest that *Mir26b* plays a protective role in MASH development and shows the therapeutic potential of *Mir26b* loaded LNPs.

In this study, *Apoe$^{-/-}$Mir26b$^{-/-}$* mice showed elevated hepatic cholesterol and triglyceride levels, which coincided with increased expression of *Msr1* and *Cd36*. Generally, MSR1 and CD36 play a critical role in lipid metabolism, as 75–90% of oxidized low-density lipoprotein (ox-LDL) is taken up by macrophages via these receptors (*Kunjathoor et al., 2002*). In line with our observations, previous studies already highlighted that CD36 and MSR1 play a key role in MASH development. For example, a hematopoietic deficiency of these receptors resulted in reduced hepatic inflammation (*Bieghs et al., 2010*). Furthermore, a recent study demonstrated that mice lacking MSR1 show decreased hepatic lipid levels and inflammation (*Govaere et al., 2022*). Interestingly, they also demonstrated that MSR1 induced JNK signaling (*Govaere et al., 2022*), which is also in line with our kinase activity results. Moreover, it could be demonstrated that *CD36* expression is upregulated in the livers of MASLD patients, leading to hepatic triglyceride accumulation and consequently an exacerbation of hepatic steatosis (*Pei et al., 2020*). Overall, this strongly indicates that an increased hepatic expression of *Cd36* and *Msr1* in both mice and humans causes hepatic cholesterol and triglyceride accumulation as well as inflammation, adding fuel to the notion of CD36 and MSR1 as a possible underlying mechanism through which *Mir26b* exerts its effects on MASH development.

Elevated hepatic lipid levels lead to the release of pro-inflammatory cytokines, consequently mediating liver injury and inflammation in MASH (*Bocsan et al., 2017*), which could also be observed in our study. The elevated levels of pro-inflammatory cytokines in the liver led to the increased number of Mac-1-positive cells, representing infiltrated macrophages. The production of IL-6 and TNF by macrophages in turn led to further local inflammation, also highlighted by the increased inflammatory kinase activity in the liver, thereby causing a vicious cycle of cytokine release and myeloid cell infiltration (*Arrese et al., 2016*). Interestingly, TNF has been shown to promote steatosis by altering lipid metabolism and inducing fatty acid uptake in the liver (*Endo et al., 2007*; *Kim et al., 2007*), further showing the critical link between inflammation and hyperlipidemia. Additionally, infiltrated macrophages usually form clusters, especially in areas of macrovesicular steatosis (*Daemen et al., 2021*), which was in line with our current study. Overall, these findings indicate a crosstalk between hepatic lipid levels, inflammation, and macrophage infiltration and show that *Mir26b* might play a protective role in these processes.

The third characteristic of MASH that was investigated during this study was hepatic fibrogenesis. Hepatic fibrosis is mainly caused by the activation of the fibrogenic factor TGF-β (*Katsarou et al., 2020*), which is also confirmed in the current study as *Mir26b* knockout mice displayed a higher amount of Sirius red positive area and an increased expression of *Tgfb*. Additionally, *Mmp13*, a protein responsible for extracellular matrix degradation, was upregulated in *Apoe$^{-/-}$Mir26b$^{-/-}$* mice, which is probably a secondary compensatory response to the increase in fibrosis. Besides this, MMP13 has been shown to play a less straightforward role in liver fibrosis (*Uchinami et al., 2006*). Uchinami et al. demonstrated that levels of TNF and TGF-β were suppressed in MMP13-deficient mice in an early phase of liver fibrosis, suggesting that MMP13 possibly accelerates hepatic fibrogenesis by

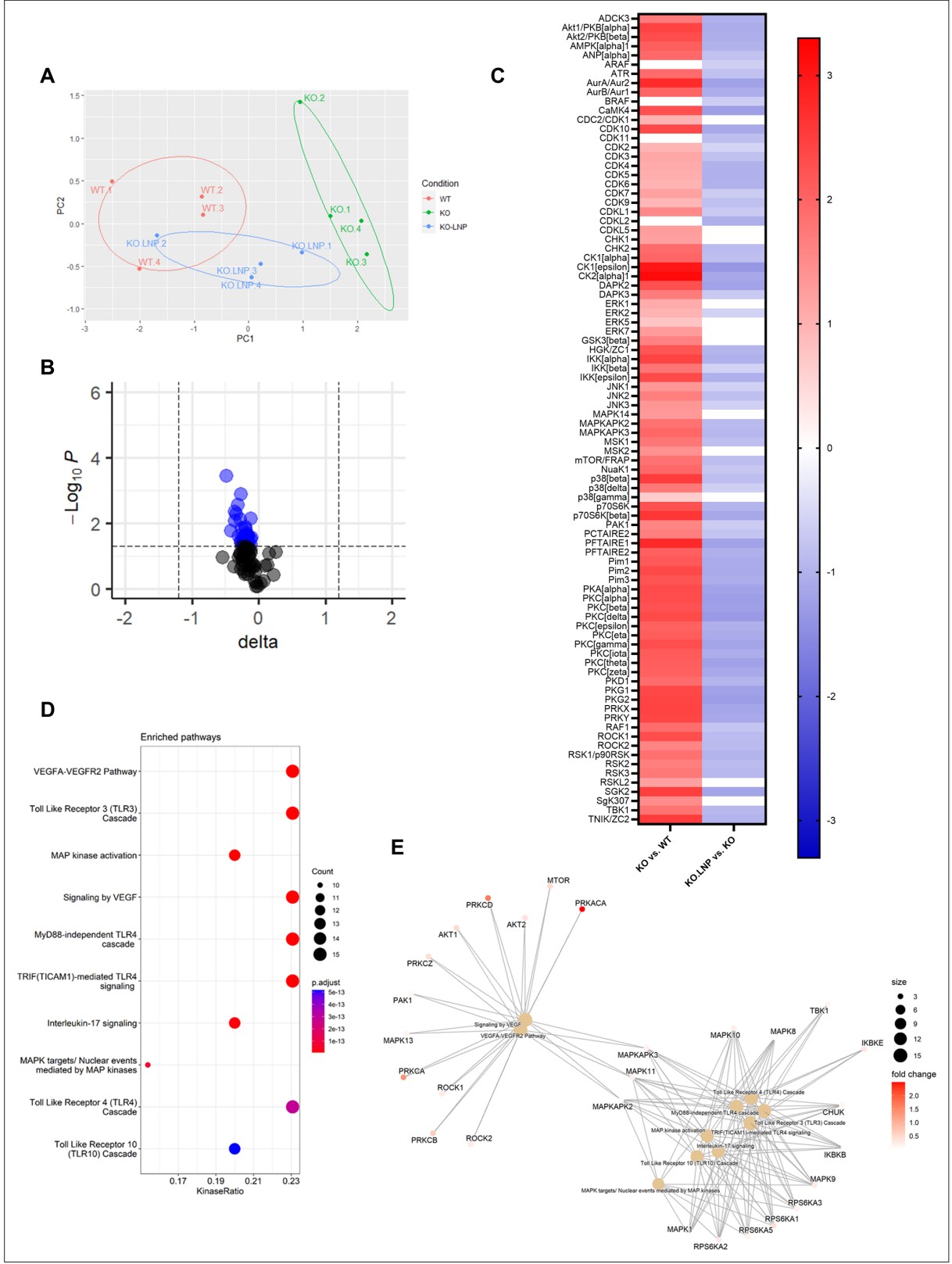

**Figure 7.** mLNP treatment of *Mir26b* knockout mice results in a decreased hepatic inflammatory kinase activity. (**A**) Principal component analysis (PCA) of phosphorylated peptides from STK array (n=4) of liver lysates from mLNP treated *Apoe$^{-/-}$Mir26b$^{-/-}$* mice (KO.LNP), *Apoe$^{-/-}$Mir26b$^{-/-}$* mice (KO) or *Apoe$^{-/-}$* mice (WT) mice. (**B**) Volcano plot visualizing fold change and p value for phosphorylated peptides from STK array. Blue dots represent significantly altered phosphopeptides. (**C**) The heatmap of significantly changed kinases is ranked based on the Median Final Score (cut-off value of 1.2). Color is

*Figure 7 continued on next page*

*Figure 7 continued*

corresponding to Median Kinase Statistic, which represents effect size and directionality (red = increased activity in KO vs. WT mice; blue = decreased activity in KO.LNP vs. KO mice; average of n=4 is shown). (**D**) Enriched pathways based on STK array. (**E**) Network diagram of the pathway enrichment analysis.

The online version of this article includes the following source data and figure supplement(s) for figure 7:

**Source data 1.** Raw data from *Figure 7*.

**Figure supplement 1.** mLNP treatment rescues the inflammatory kinase activity effect of *Mir26b* knockout.

**Figure supplement 1—source data 1.** Raw data from *Figure 7—figure supplement 1*.

mediating inflammation. Overall, this indicates that *Mir26b* plays a protective role in hepatic fibrogenesis, possibly due to modulating TGF-β and MMP13 levels.

Finally, to study the therapeutic potential of *Mir26b*, we injected whole-body knockout mice with *Mir26b* mimic-loaded LNPs. LNP-based therapies have emerged over the last few years and have been extensively studied and even already used in clinical settings (*Peters et al., 2021*), underlining the translatability and potential of our study. Interestingly, the LNP treatment not only showed significant results on hepatic lipid metabolism but also reversed the effects on inflammatory kinase activity, particularly of pathways mediated by key regulators like JNKs, MAPKs, and ERKs. However, the LNP treatment did not seem to affect the inflammatory cytokine, cellular leukocyte influx or fibrosis in the

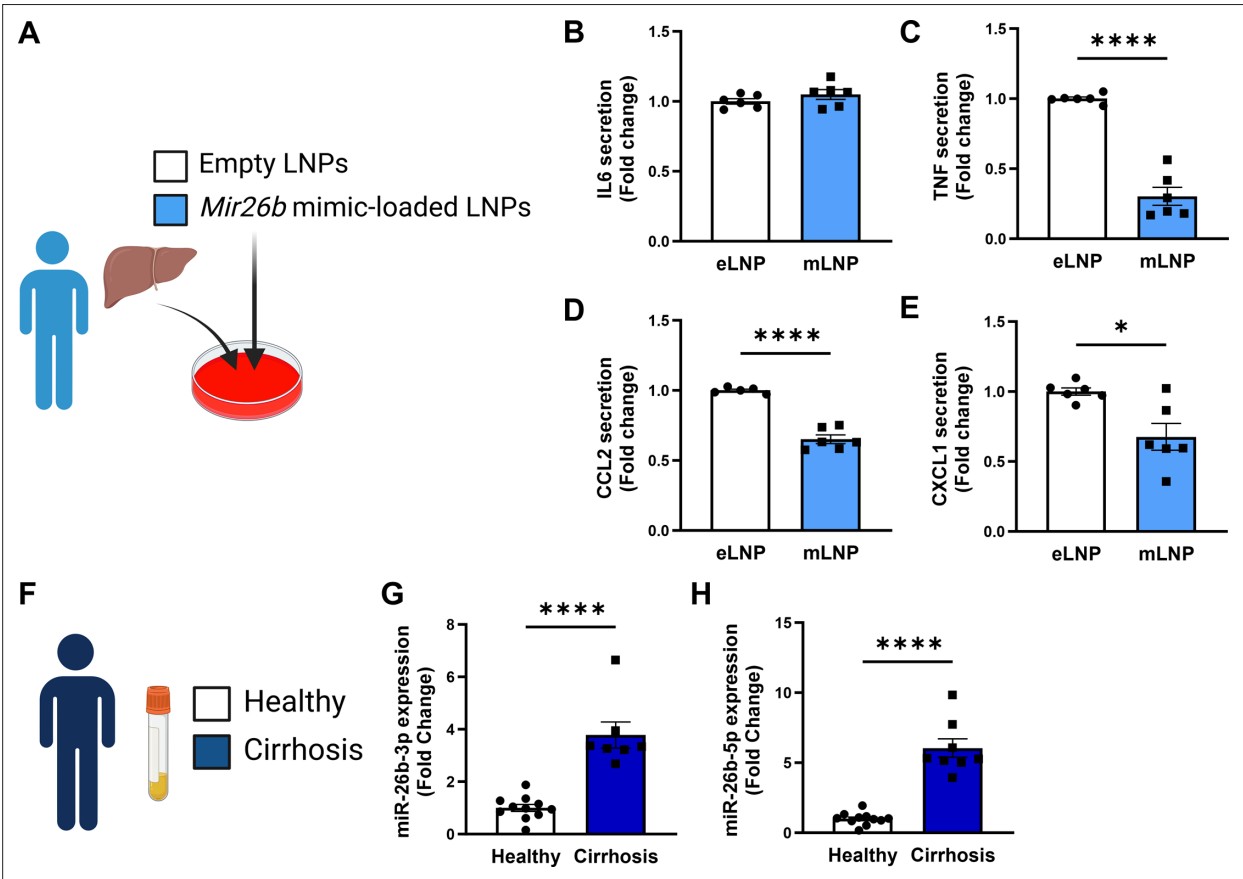

**Figure 8.** *Mir26b*-loaded LNPs have anti-inflammatory effects on human liver slices and *Mir26b* plasma levels are significantly increased in patients with liver cirrhosis. (**A**) Schematic overview of the experimental approach. (**B–E**) Cytokine levels of (**B**) IL-6, (**C**) TNF, (**D**) CCL2, and (**E**) CXCL1 measured in the supernatant of human precision-cut liver slices after 24 hr (for IL-6/TNF) or 48 hr (for CCL2/CXCL1) incubation with mLNPs or eLNPs (3 individual donors, cultured in duplicates). (**F–G**) Plasma was isolated from patients with liver cirrhosis or healthy volunteers (**F**) and *Mir26b*-3p (**G**) and *Mir26b*-5p (**H**) plasma levels were measured. *p<0.05; ****p<0.0001. n=8–11 patients per group. Panels **A** and **F** were created using BioRender.com.

The online version of this article includes the following source data for figure 8:

**Source data 1.** Source data contains the raw data from *Figure 8*.

liver, suggesting that the treatment might be to short to see such functional effects on later disease stages. Nonetheless, a note of caution is the fact that we injected the LNPs simultaneously with the diet, providing a more preventative approach as opposed to a curative therapy. Furthermore, we demonstrated that *Mir26b* loaded LNPs have anti-inflammatory effects on human precision-cut liver slices, which, combined with the suppressed levels of cholesterol and hepatic lipids in our mouse model, underline its potential as an exciting therapeutic option.

Besides the in vivo studies, we also examined the expression levels of *Mir26b* in the plasma of liver cirrhosis patients. Compared to healthy subjects, both *Mir26b*-3p and -5p were expressed higher in patients with liver cirrhosis. This is in contrast to our mouse model in which higher expression levels led to a more beneficial phenotype. Important to note is that the levels of *Mir26b* in the plasma of these patients do not prove causality. The increase in *Mir26b* expression could be due to a mechanism in which the body attempts to rescue the diseased phenotype. This would make the MicroRNA an interesting target as a biomarker though. However, more research should be conducted as the size of the human study is rather limited and only focuses on one of the last stages of MASLD.

Collectively, these results show that *Mir26b* plays a protective role in the development of MASH by exerting effects on hepatic lipid metabolism, inflammation, and fibrosis. Future research should focus on the further clinical translation of our results by evaluating the effects of *Mir26b* loaded LNPs on various human tissues. Overall, we show here thought-provoking results on the role of *Mir26b* in MASH development and highlight this MicroRNA as a promising therapeutic target, providing a solid base for exciting future research.

## Materials and methods

**Key resources table**

| Reagent type (species) or resource | Designation | Source or reference | Identifiers | Additional information |
|---|---|---|---|---|
| Strain, strain background (*Mus musculus*) | *Apoe$^{-/-}$Mir26b$^{-/-}$* | *van der Vorst et al., 2021* | - | - |
| Strain, strain background (*M. musculus*) | *Apoe$^{-/-}$* | Jackson | Strain #:002052 | - |
| Strain, strain background (*M. musculus*) | *Apoe$^{-/-}$Mir26b$^{fl/fl}$* | Generated in house | - | - |
| Strain, strain background (*M. musculus*) | *Apoe$^{-/-}$Mir26b$^{fl/fl}$ Lyz2$^{Cre+}$* | Generated in house | - | - |
| Other | Western-type diet | Sniff | TD88137 | Mouse diet |

### Animals

*Apoe$^{-/-}$Mir26b$^{-/-}$* mice were generated as described in *van der Vorst et al., 2021*. *Apoe$^{-/-}$* mice were used as control and all mice were on C57BL/6 J background for more than 10 generations. Both male and female mice were used for this study. Furthermore, myeloid lineage-specific *Mir26b$^{fl/fl}$* mice on an *Apoe$^{-/-}$* background were generated by crossing *Apoe$^{-/-}$Mir26b$^{fl/fl}$* (generated in house) with *Lyz2$^{Cre+}$* transgenic mice (Jackson; strain #004781) to form *Apoe$^{-/-}$Mir26b$^{fl/fl}$Lyz2$^{Cre+}$* mice. *Apoe$^{-/-}$Mir26b$^{fl/fl}$Lyz2$^{Cre-}$* mice were used as control. Starting at 8 weeks of age, the mice were fed a Western-type diet (WTD) containing 21% fat and 0.20% cholesterol (Sniff TD88137) for 4 or 12 weeks after which the mice were euthanized by intraperitoneal injection of ketamine (0.1 mg/g body weight) and xylazine (0.02 mg/g body weight). The blood was collected, and livers were harvested and snap-frozen using liquid nitrogen. The livers were embedded in Tissue-Tek O.C.T. compound (Sakura) and cryosectioned in 5 µm cuts and mounted on glass slides. For metabolic measurements and further analysis, 100 µm tissue sections were collected in 2 ml reaction tubes. All animal studies performed were approved by the local ethical committee (Landesamt für Natur, Umwelt und Verbraucherschutz Nordrhein-Westfalen, Germany, approval number 81–02.04.2019 .A363 and Regierung von Ober-bayern, approval number 55.2-1-54-253245-2015).

### Lipid nanoparticle production

The ionizable cationic lipid DLin-MC3-DMA (Hycultec GmbH, Beutelsbach, Germany) was combined with DSPC (1,2-distearoyl-sn-glycero-3-phosphorylcholine, Avanti), Cholesterol (Sigma) and DMG-PEG 2000 (Avanti) in absolute ethanol at molar ratios of 50:10:38.5:1.5 and a final lipid concentration of 50 mM. The aqueous phase was prepared by dissolving a combination (1:1) of *Mir26b*-3P and –5 P

in sterile 100 mM acetate buffer (pH 4), to achieve a 0.17 µg/µL solution. For LNP formulation, the ethanol and aqueous phase were injected into a microfluidic herringbone mixer (Microfluidic Chip-Shop, Jena, Germany) via syringe pumps at a flow rate ratio of 1:3, respectively, with a total flow rate of 4 mL/min to obtain an N/P ratio of 4. Generated LNPs were diluted with PBS to a concentration of ethanol below 2%, followed by concentrating via ultra-centrifugation ($3000 \times g$) using 10 kDa Amicon Ultrafilitration units (Sigma) and were finally dialyzed against PBS using a dialysis membrane with 30 kDa MWCO.

To characterize the particle size and surface charge, samples were diluted with PBS and equilibrated for 15 min at 20 °C before analysis. Particle sizes and polydispersity were determined by dynamic light scattering and Zetapotential measurements were carried out both, using a Zetasizer Nano ZS (Malvern Instruments Ltd).

The encapsulation efficiency of RNA of prepared LNPs was determined using a Quant-iT RiboGreen assay according to the instructions of the manufacturer (Thermo Fisher Scientific). Samples were analyzed by fluorescence quantification on a microplate reader (Cytation 3, BioTek Instruments Inc). The encapsulation efficiency was calculated as the difference between the total RNA and the non-encapsulated RNA, divided by the total RNA (x100%). The dosing of RNA-LNPs was based on the Ribogreen assay result.

## LNP injections

LNPs were produced as described above LNPs containing 2 mg/kg RNA were administered via intraperitoneal injection every 3 days. Empty LNPs (eLNPs) and LNPs loaded with murine *Mir26b*-3p (Duplex Sequences: 5'-/5Phos/rCrCrUrGrUrUrCrUrCrCrArUrUrArCrUrUrGrGmCmUrC-3' and 5'-mGmCrC-mArAmGrUmArAmUrGmGrAmGrAmArCmArGG-3') and *Mir26b*-5p (Duplex Sequences: 5'-/5Phos/rUrUrCrArArGrUrArArUrUrCrArGrGrArUrAmGmGrU-3' and 5'-mCmUrAmUrCmCrUmGrAmArUmUr-AmCrUmUrGmAA-3') mimics (IDT) were injected into *Apoe$^{-/-}$miRNA-26b$^{-/-}$* mice to determine potential therapeutic effects.

## RNA isolation and quantitative polymerase chain reaction

One frozen liver piece of 25 mg per mouse was homogenized in a closed tube with glass beads and 1 mL Qiazol by using the Tissue Lyser (QIAGEN) for 5 min at 50 Hz. RNA was isolated using the miRNeasy mini Kit per the manufacturer's protocol (QIAGEN). Following RNA isolation, cDNA was made from 500 ng total RNA by adding 1 µl Oligo (dT)-Primer (Eurofins Genomics). Secondary RNA structure was denaturalized at 70 °C for 5 min after which the samples were briefly cooled on ice to allow primer annealing. Subsequently, M-MLV Reverse Transcriptase, M-MLV RT 5 x Buffer, and dNTP Mix (Promega) were added, and cDNA synthesis was completed by incubation for 1 hr at 37 °C. Quantitative polymerase chain reaction (qPCR) determined the relative gene expression levels using primer sequences listed in *Table 1*. For the qPCR reaction, 10 ng of cDNA template was used, to which 1 X PowerUp SYBR Green Master Mix (Thermo Fisher) and primers (Eurofins Genomics) were added. PCR cycling was performed on QuantStudio 3 Real-Time PCR system (Thermo Fisher) with the following

**Table 1.** Primer sequences for genes measured with qPCR.

| Primer | Sequence in 5'–3'- direction | |
| --- | --- | --- |
| | Forward | Reverse |
| *Abca1* | CCCAGAGCAAAAAGCGACTC | GGTCATCATCACTTTGGTCCTTG |
| *Acat2* | ACCAATTCCAGCCATAAAGCA | GGTTTAATCCAAGTTCTTTAGCTATTGC |
| *Acta2* | ACGAACGCTTCCGCTGC | GATGCCCGCTGACTCCAT |
| *Cd36* | GCCAAGCTATTGCGACATGA | AAAAGAATCTCAATGTCCGAGACTTT |
| *Cyclophilin* | TTCCTCCTTTCACAGAATTATTCCA | CCGCCAGTGCCATTATGG |
| *Mmp13* | ACAAAGATTATCCCCGCCTCATA | CACAATGCGATTACTCCAGATACTG |
| *Msr1* | CATACAGAAACACTGCATGTCAGAGT | TTCTGCTGATACTTTGTACACACGTT |
| *Tgfb* | GCCCTTCCTGCTCCTCATG | CCGCACACAGCAGTTCTTCTC |

conditions: 50 °C for 2 min for 1 cycle (UDG activation); 95 °C for 2 min for 1 cycle (Dual-Lock DNA polymerase); and 95 °C for 15 s (Denature), 58 °C for 15 s (Anneal) and 72 °C for 1 min (Extend) for 40 cycles. The reference gene *Cyclophilin* was used for normalization.

According to the manufacturer's instructions, RNA was isolated for the *Mir26b* PCR analysis using the miRNeasy serum/plasma kit (QIAGEN). Subsequently, cDNA was generated using the TaqMan MicroRNA Reverse Transcription Kit (Thermo Fisher) according to the manufacturer's instructions using 10 ng of total RNA. The relative gene expression levels were determined by qPCR using TaqMan MicroRNA-assays (Thermo Fisher) for *U6* (Assay-ID: 001973), *Mir26b*-3p (Assay-ID: 000407) or *Mir26b*-5p (Assay-ID: 002444). For the qPCR reaction, 0.5 ng of cDNA template was used to which TaqMan gene expression master mix (Thermo Fisher) and above-described primers were added. PCR cycling was performed on QuantStudio 3 Real-Time PCR system (Thermo Fisher) with the following conditions: 50 °C for 2 min for 1 cycle (UDG activation); 95 °C for 10 min for 1 cycle (Dual-Lock DNA polymerase); and 95 °C for 15 s (Denature), 60 °C for 60 s (Anneal/Extend) for 40 cycles. Expression of *U6* was used for normalization.

## Protein isolation

One frozen liver piece of 25 mg per mouse was homogenized in 0.5 ml SET buffer (sucrose 250 mmol/L, EDTA 2 mmol/L, TRIS 10 mmol/L) by vortexing briefly. Cell destruction was completed by 2 freeze-thaw cycles with liquid nitrogen and subsequently passing the sample through a 27-gauge needle for three times. After one last freeze-thaw cycle, the total protein content was measured using the NanoDrop One Microvolume UV-Vis Spectrophotometer (Thermo Fisher Scientific).

## Lipid analysis

Cholesterol and triglyceride levels were quantified in liver protein lysates and EDTA-plasma using enzymatic colorimetric assays (c.f.a.s. cobas, Roche Diagnostics) according to the manufacturer's protocol. Absorbance was measured at 510 nm with the microplate reader infinite M200 (Tecan).

## Enzyme-linked immunosorbent assay

Mouse TNF, interleukin-6 (IL-6), CC-chemokine ligand 2 (CCL2), and C-X-C Motif Chemokine Ligand 1 (CXCL1) levels were measured in liver protein lysates and EDTA-plasma by ELISA (Thermo Fisher) according to the manufacturer's instructions. Absorbance was measured at 450 nm with wavelength subtraction at 570 nm using the microplate reader infinite M200 (Tecan).

According to the manufacturer's instructions, human TNF, IL-6, CCL2, and CXCL1 levels were measured in the supernatant of human precision-cut liver slices using ELISA (Thermo Fisher). Absorbance was measured at 450 nm with wavelength subtraction at 570 nm using the microplate reader infinite M200 (Tecan).

## Western blotting

One 100 µm liver section was lysed for 15 min on ice using M-PER Mammalian Extraction Buffer containing Halt Phosphatase Inhibitor and EDTA-free Halt Protease Inhibitor Cocktail (1:100 each; Thermo Fisher Scientific). Lysates were centrifuged for 15 min at $16000 \times g$ at +4 °C in a pre-cooled centrifuge. Protein quantification was performed with a NanoDrop One Microvolume UV-Vis Spectrophotometer (Thermo Fisher Scientific). An equal amount of protein from each sample was resolved by 10% SDS–polyacrylamide gel electrophoresis, transferred to nitrocellulose membranes, and blocked with 5% bovine serum albumin (BSA) for 1 hr at room temperature. Anti-CD36 antibody (1:1000; Cell signalling), anti-MSR1 antibody (1:1000; Abcam), and anti-β-actin (1:1000; Cell signalling) were used as primary antibodies. The blots were incubated overnight at 4 °C. An anti-rabbit antibody (1:1000; Cell signalling) was used and incubated for 1 hr at room temperature. Immunoreactive bands were visualized via enhanced chemiluminescence in a ChemiDoc Imager (Bio-Rad), and densitometry was performed using Image J. β-actin was used for normalization.

## Oil-red-O staining

Liver sections were prepared as described above. Following 30 min drying to the air, the cryo sections were fixed with 3.5% formaldehyde for 30 min at room temperature. Then the liver sections were stained with Oil-red-O (Sigma-Aldrich) for 1 hr and counterstained with Mayer's heamlum solution

(Merck) for 30 s. After mounting the slides with glycerin jelly, images were acquired with an automated upright microscope (Leica microsystems), and the lipid content in the livers was quantified using ImageJ Fiji software (Laboratory for Optical and Computational Instrumentation, University of Wisconsin-Madison, Madison, Wisconsin, United States). All analyses were performed in a blinded manner.

## Immunofluorescent stainings

Several immunofluorescent stainings were performed to visualize inflammation and leukocyte infiltration. The cryo-sectioned livers were first air-dried for 5 min at room temperature and subsequently fixed with ice-cold acetone for 10 min. Tissue sections were blocked with 1% bovine serum albumin (BSA) and 0.03% normal horse serum blocking solution (Vector) in 1 X PBS for 1 hr. Neutrophils, resident macrophages, infiltrating neutrophils and macrophages, and T-cells were visualized by staining with anti-mouse Ly6G (Biolegend, dilution 1:100), anti-mouse CD68 (Biolegend, dilution 1:250), anti-mouse Mac-1 (R&D Systems, dilution 1:100) or anti-mouse CD3 (Abcam, dilution 1:100), respectively, overnight at 4 C. Liver sections were incubated with secondary antibodies Cy3-conjugated donkey anti-rat IgG (Jackson ImmunoResearch, dilution 1:300) or Cy3-conjugated donkey anti-rabbit IgG (Jackson ImmunoResearch, dilution 1:300) for 30 min at room temperature after which nuclei were stained with Hoechst (Thermo Fisher, dilution 1:10,000) for 10 min at room temperature. Following the staining, the sections were mounted with Immuno-Mount (Thermo Fisher), and images were acquired using an inverted microscope Dmi8 (Leica microsystems). The number of Ly6-G, CD68, Mac-1, and CD3 positive immune cells was counted with the ImageJ Fiji software. All analyses were performed in a blinded manner.

## Picrosirius red staining

Liver fibrosis was visualized by a Picrosirius red staining. First, the liver sections were fixed for 2 hr in 10% formalin. This was followed by a 90-min incubation with 0.1% Sirius Red (Polysciences) in 1% saturated picric acid solution (Applichem). Slides were subsequently incubated in 0.01 N HCl for 2 min and dehydrated using an ethanol range. After incubation in xylol, the slides were mounted with Vitro-Clud (R.Langenbrinck). Images were acquired using an automated upright microscope (Leica microsystems), after which the sirius red positive area was analyzed and calculated in each liver section by using ImageJ Fiji. All analyses were performed in a blinded manner.

## Kinase activity profiling

STK profiles were determined using the PamChip Ser/Thr Kinase assay (STK; PamGene International, ´s-Hertogenbosch, The Netherlands). Each STK-PamChip array contains 144 individual phospho-site(s) that are peptide sequences derived from substrates for STKs. One 100 µm liver section was lysed for 15 min on ice using M-PER Mammalian Extraction Buffer containing Halt Phosphatase Inhibitor and EDTA-free Halt Protease Inhibitor Cocktail (1:100 each; Thermo Fisher Scientific). Lysates were centrifuged for 15 min at $16000 \times g$ at +4 °C in a pre-cooled centrifuge. Protein quantification was performed with a NanoDrop One Microvolume UV-Vis Spectrophotometer (Thermo Fisher Scientific).

For the STK assay, 2.0 µg of protein and 400 µM ATP were applied per array (n=4 per condition) together with an antibody mix to detect the phosphorylated Ser/Thr. After incubation for an hour (30 °C) where the sample is pumped back and forth through the porous material to maximize binding kinetics and minimize assay time, a 2nd FITC-conjugated antibody is used to detect the phosphorylation signal. Imaging was done using an LED imaging system. The spot intensity at each time point was quantified (and corrected for local background) using the BioNavigator software version 6.3 (PamGene International, 's-Hertogenbosch, The Netherlands). Upstream Kinase Analysis (UKA), a functional scoring method (PamGene) was used to rank kinases based on combined specificity scores (based on peptides linked to a kinase, derived from six databases) and sensitivity scores (based on treatment-control differences).

For the peptides, a principal component analysis (PCA) was performed with the help of the R package stats V4.3.1, depicting the singular peptide decomposition and examining the covariances/correlations between the samples. The R package gglot2 v3.4.2 was used to visualize the results. The distribution of the phosphorylated peptides is shown in the volcano plots created with the R package EnhancedVolcano v1.18.0. Blue dots depict peptides with an adjusted p-value <0.05.

For kinases, the median final score of the kinase with a score >1.2 and with an adjusted P value for multiple comparisons by the false discovery rate (FDR) of <0.05 are depicted in a heatmap. The R package disease ontology semantic and enrichment analysis (DOSE; *Yu et al., 2015*) was utilized to analyze the enriched pathways depicting the biological complexities in which these kinases correlate with multiple annotation categories, which was visualized in a network plot with the help of the R package Reactome Pathway Analysis (ReactomePA) v1.44.0 (*Yu and He, 2016*).

## Precision-cut liver slices

Small human liver wedges of an equivalent size of approximately 10 g were collected from three human donors following partial resection or when livers were unsuitable for transplantation. The study was approved by the Medical Ethical Committee of the University Medical Centre Groningen (UMCG), according to Dutch legislation and the Code of Conduct for dealing responsibly with human tissue in the context of health research, refraining the need for written consent for 'further use' of coded-anonymous human tissue. The procedures were carried out in accordance with the experimental protocols approved by the Medical Ethical Committee of the UMCG. Liver tissue was stored in University of Wisconsin preservation solution (UW, 4 °C). The total cold ischemic time was between 3 and 29 hr. Slice viability for each donor liver was tested after 1 hr of culture by checking ATP production as previously described (*de Graaf et al., 2010*). Slices were cultured in GFIPO medium (*Simon et al., 2023*; 36 mM Glucose, 5 mM Fructose, 1 nM Insulin, 480 μM Oleic acid, 240 μM Palmitic acid) and cultured for 24 hr and 48 hr in the presence of empty LNPs (eLNPs) or LNPs loaded with murine *Mir26b*-3p (Duplex Sequences: 5'-/5Phos/rCrCrUrGrUrUrCrUrCrCrArUrUrArCrUrUrGrGmCmUrC-3' and 5'-mGmCrCmArAmGrUmArAmUrGmGrAmGrAmArCmArGG-3') and *Mir26b*-5p (Duplex Sequences: 5'-/5Phos/rUrUrCrArArGrUrArArUrUrCrArGrGrArUrAmGmGrU-3' and 5'-mCmUrAmUrC-mCrUmGrAmArUmUrAmCrUmUrGmAA-3') mimics (IDT). The medium was refreshed every 24 hr. The slices were collected to check the viability of the slices by measuring ATP levels and supernatant was collected for further analysis.

## *Mir26b* expression in patient cohort

All patients were prospectively recruited in the Department of Medicine II (Saarland University Medical Center, Homburg, Germany) between December 2021 and March 2023. Included patients were adults and had either type 1 or type 2 diabetes. Alcohol consumption above the National Institute on Alcohol Abuse and Alcoholism's (NIAAA) definition of chronic alcohol use (four drinks or more on any day or 14 drinks per week for men or three drinks or more on any day or 7 drinks per week for women) was regarded as exclusion criterium. Serum and EDTA blood samples were collected from fasted patients. Genomic DNA was isolated from EDTA anticoagulated blood according to the membrane-based QIAamp DNA extraction protocol (QIAGEN, Hilden, Germany). The common genetic variants involved known to modulate the risk of fatty liver, namely *PNPLA3*, *TM6SF2*, *MBOAT7*, *SERPINA*, *HSD17B13*, and *MTARC1*, were genotyped using a solution-phase hybridization reaction with 5'-nuclease and fluorescence detection. Transient elastography (TE) and controlled attenuation parameter (CAP) were performed to non-invasively quantify liver fibrosis and steatosis, respectively. Cirrhosis was defined by TE greater or equal 15 kPa, and fibrosis F0 was defined by TE below 6.5 kPa (*Mózes et al., 2022*; *Siddiqui et al., 2019*).

RNA was isolated from 100 μl serum, using the miRNeasy serum/plasma kit (QIAGEN), according to the manufacturer's instructions. *C. elegans Mir39* miRNA mimic was added as spike-in control. Subsequently, cDNA was generated using the TaqMan MicroRNA Reverse Transcription Kit (Thermo Fisher) according to the manufacturer's instructions using 10 ng of total RNA. The relative gene expression levels were determined by qPCR using TaqMan MicroRNA-assays (Thermo Fisher) for *Mir39* (Assay-ID: 000200), *Mir26b*-3p (Assay-ID: 000407) or *Mir26b*-5p (Assay-ID: 002444). For the qPCR reaction, 0.5 ng of cDNA template was used to which TaqMan gene expression master mix (Thermo Fisher) and above-described primers were added. PCR cycling was performed on Quant-Studio 3 Real-Time PCR system (Thermo Fisher) with the following conditions: 50 °C for 2 min for 1 cycle (UDG activation); 95 °C for 10 min for 1 cycle (Dual-Lock DNA polymerase); and 95 °C for 15 s (Denature), 60 °C for 60 s (Anneal/Extend) for 40 cycles. Expression of *Mir39* as spike-in control was used for normalization.

## Statistics

Statistical analysis was performed using GraphPad Prism version 9.1.1 (GraphPad Software, Inc, San Diego, CA, USA). Outliers were identified using the ROUT = 1 method after which normality was tested via the D'Agostino-Pearson and Shapiro-Wilk normality test. Significance was tested using either Welch's t-test for normally distributed data or Mann-Whitney U test for non-normally distributed data. All data are expressed as mean ± SEM and results of <0.05 for the p-value were considered statistically significant.

All authors had access to the study data and have reviewed and approved the final manuscript.

## Materials availability statement

All data generated or analysed during this study are included in the manuscript and supporting files; source data files have been provided for all figures.

## Acknowledgements

This research was funded by grants from the Interdisciplinary Center for Clinical Research within the faculty of Medicine at the RWTH Aachen University, NWO-ZonMw Veni (91619053), by the German Research Foundation (DFG SFB/TRR219; Project-ID 322900939; Subproject M07), by the Corona Foundation (S199/10084/2021) and the Fritz Thyssen Stiftung (Grant No. 10.20.2.043MN) to EPCvdV; by the Austrian Science Fund (FWF) (ZK81-B, P36774-B) to TH; by the DFG (BA6226/2-1, BA6226/2-3), the Wilhelm Sander Foundation (Grant No. 2018.129.1), the COST Action Mye-InfoBank 476 (CA20117), a BMBF grant (16LW0143, Mamma-Explant), the state of NRW (ZM-1-27B, NovoKolon), and an ERS grant from RWTH University (G:(DE-82)EXS-SF-OPSF732) to MB; by the Swiss National Foundation project ID 310030-197655 to YD; by a CSC stipend (ID: 202008320329) to CL; by the 'Deutsche Forschungsgemeinschaft` (DFG, German Research Foundation) by the Transregional Collaborative Research Centre (SFB TRR219, Project-ID 322900939); and CRC 1382 (Project-ID: 403224013) to JJ; by the BMBF in the framework of the Cluster4Future program (Cluster for Nucleic Acid Therapeutics Munich, CNATM) to CW. Human precision-cut liver slices by "Meer Kennis met Minder Dieren" under project number 114022505, which is partly financed by the ZonMw program of the Dutch Scientific Organization and Proefdiervrij to PO. Schematic overview figures are created in BioRender.

## Additional information

### Funding

| Funder | Grant reference number | Author |
| --- | --- | --- |
| Interdisciplinary Center for Clinical Research within the faculty of Medicine at the RWTH Aachen University | | Emiel van der Vorst |
| NWO-ZonMw Veni | 91619053 | Emiel van der Vorst |
| German Research Foundation | DFG SFB/TRR219, Project ID 322900939, Subproject M07 | Emiel van der Vorst |
| Corona-Stiftung | S199/10084/2021 | Emiel van der Vorst |
| Fritz Thyssen Stiftung | 10.20.2.043MN | Emiel van der Vorst |
| Austrian Science Fund | ZK81-B | Tim Hendrikx |
| Austrian Science Fund | 10.55776/P36774 | Tim Hendrikx |
| German Research Foundation | BA6226/2-1 | Matthias Bartneck |
| German Research Foundation | BA6226/2-3 | Matthias Bartneck |

| Funder | Grant reference number | Author |
|---|---|---|
| Wilhelm Sander Foundation | 2018.129.1 | Matthias Bartneck |
| COST Action Mye-InfoBank 476 | CA20117 | Matthias Bartneck |
| Bundesministerium für Bildung und Forschung | 16LW0143 Mamma-Explant | Matthias Bartneck |
| State of NRW | ZM-1-27B NovoKolon | Matthias Bartneck |
| RWTH Aachen University | ERS grant G:(DE-82)EXS-SF-OPSF732 | Matthias Bartneck |
| Swiss National Science Foundation | Project ID 310030-197655 | Yvonne Döring |
| China Scholarship Council | CSC stipend 202008320329 | Cheng Lin |
| German Research Foundation | SFB TRR219, Project ID 322900939 | Joachim Jankowski |
| German Research Foundation | CRC 1382, Project-ID: 403224013 | Joachim Jankowski |
| Bundesministerium für Bildung und Forschung | Cluster4Future: Cluster for Nucleic Acid Therapeutics Munich (CNATM) | Christian Weber |
| ZonMw | Meer Kennis met Minder Dieren 114022505 | Peter Olinga |

The funders had no role in study design, data collection and interpretation, or the decision to submit the work for publication.

## Author contributions

Linsey Peters, Data curation, Formal analysis, Writing - original draft; Leonida Rakateli, Data curation, Formal analysis, Investigation; Rosanna Huchzermeier, Andrea Bonnin-Marquez, Sanne L Maas, Yana Geng, Alan Gorter, Marion Gijbels, Data curation, Formal analysis; Cheng Lin, Alexander Jans, Data curation, Methodology; Sander Rensen, Peter Olinga, Tim Hendrikx, Marcin Krawczyk, Malvina Brisbois, Christian Weber, Ronit Shiri-Sverdlov, Tom Houben, Resources; Joachim Jankowski, Erik AL Biessen, Resources, Supervision; Kiril Bidzhekov, Resources, Formal analysis; Yvonne Döring, Resources, Supervision, Writing – review and editing; Matthias Bartneck, Resources, Methodology, Writing – review and editing; Emiel van der Vorst, Conceptualization, Data curation, Formal analysis, Supervision, Funding acquisition, Project administration, Writing – review and editing

## Author ORCIDs
Ronit Shiri-Sverdlov ⓘ https://orcid.org/0000-0002-6736-7814
Emiel van der Vorst ⓘ https://orcid.org/0000-0001-5771-6278

## Ethics

The study was approved by the Medical Ethical Committee of the University Medical Centre Groningen (UMCG), according to Dutch legislation and the Code of Conduct for dealing responsibly with human tissue in the context of health research, refraining the need for written consent for 'further use' of coded-anonymous human tissue. The procedures were carried out in accordance with the experimental protocols approved by the Medical Ethical Committee of the UMCG.

All animal studies performed were approved by the local ethical committee (Landesamt für Natur, Umwelt und Verbraucherschutz Nordrhein-Westfalen, Germany, approval number 81-02.04.2019. A363 and Regierung von Oberbayern, approval number 55.2-1-54-253245-2015).

Reviewer #1 (Public review): https://doi.org/10.7554/eLife.97165.3.sa1
Reviewer #2 (Public review): https://doi.org/10.7554/eLife.97165.3.sa2
Author response https://doi.org/10.7554/eLife.97165.3.sa3

# Additional files

## Supplementary files
MDAR checklist

## Data availability
All data generated or analysed during this study are included in the manuscript and supporting files; source data files have been provided for all figures.

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
