## [Editor Report · eLife Assessment]

This study presents **valuable** insights into the involvement of miR-26b in the progression of metabolic dysfunction-associated steatohepatitis (MASH). The delivery of microRNA-containing nanoparticles to reduce MASH severity has practical implications as a therapeutic strategy. The authors use two sets of transgenic mouse models, conducted kinase activity profiling of mouse liver samples, and supplemented their findings with additional experiments on human liver and plasma, providing **solid** support for their findings.

---

## [Referee Report · Reviewer #1 (Public review)]

Based on previous publications suggesting a potential role for miR-26b in the pathogenesis of metabolic dysfunction-associated steatohepatitis (MASH), the researchers aim to clarify its function in hepatic health and explore the therapeutical potential of lipid nanoparticles (LNPs) to treat this condition. First, they employed both whole-body and myeloid cell-specific miR-26b KO mice and observed elevated hepatic steatosis features in these mice compared to WT controls when subjected to WTD. Moreover, livers from whole-body miR-26b KO mice also displayed increased levels of inflammation and fibrosis markers. Kinase activity profiling analyses revealed distinct alterations, particularly in kinases associated with inflammatory pathways, in these samples. Treatment with LNPs containing miR-26b mimics restored lipid metabolism and kinase activity in these animals. Finally, similar anti-inflammatory effects were observed in the livers of individuals with cirrhosis, whereas elevated miR-26b levels were found in the plasma of these patients in comparison with healthy control. Overall, the authors conclude that miR-26b plays a protective role in MASH and that its delivery via LNPs efficiently mitigates MASH development.

The study has some strengths, most notably, its employ of a combination of animal models, analyses of potential underlying mechanisms, as well as innovative treatment delivery methods with significant promise. However, it also presents certain weaknesses that could be improved. The precise role of miR-26b in a human context remains elusive, hindering direct translation to clinical practice.

Comments on revised version:

Some of the recommendations provided by this Reviewer in the first version of the manuscript have been successfully addressed in the revision. However, others, particularly those related to human translation, remain unresolved due to the lack of additional samples for analysis. Since the revised title now indicates that the mechanisms described were primarily observed in mice, it seems reasonable to defer addressing this issue to future studies.

---

## [Referee Report · Reviewer #2 (Public review)]

Summary:

This manuscript by Peters, Rakateli et al. aims to characterize the contribution of miR-26b in a mouse model of metabolic dysfunction-associated steatohepatitis (MASH) generated by Western-type diet on background of Apoe knock-out. In addition, the authors provide a rescue of the miR-26b using lipid nanoparticles (LNPs), with potential therapeutic implications. In addition, the authors provide useful insights on the role of macrophages and some validation of the effect of miR-26b LNPs on human liver samples.

Strengths:

The authors provide a well designed mouse model, that aims to characterize the role of miR-26b in a mouse model of metabolic dysfunction-associated steatohepatitis (MASH) generated by Western-type diet on background of Apoe knock-out. The rescue of the phenotypes associated with the model used using miR-26b using lipid nanoparticles (LNPs) provides an interesting avenue to novel potential therapeutic avenues.

Weaknesses:

Although the authors provide a new and interesting avenue to understand the role of miR-26b in MASH, the study needs some additional validations and mechanistic insights in order to strengthen the authors' conclusions.

(1) Analysis the expression of miRNAs based on miRNA-seq of human samples (see https://ccb-compute.cs.uni-saarland.de/isomirdb/mirnas) suggests that miR-26b-5p is highly abundant both on liver and blood. It seems hard to reconcile that despite miRNA abundance being similar on both tissues, the physiological effects claimed by the authors in Figure 2 come exclusively from the myeloid (macrophages).

- Thanks for the clarification provided on your revised version of the manuscript

(2) Similarly, the miRNA-seq expression from isomirdb suggests also that expression of miR-26a-5p is indeed 4-fold higher than miR-26b-5p both in liver and blood. Since both miRNAs share the same seed sequence, and most of the supplemental regions (only 2 nt difference), their endogenous targets must be highly overlapped. It would be interesting to know whether deletion of miR-26b is somehow compensated by increased expression of miR-26a-5p loci. That would suggest that the model is rather a depletion of miR-26.

UUCAAGUAAUUCAGGAUAGGU mmu-miR-26b-5p mature miRNA

UUCAAGUAAUCCAGGAUAGGCU mmu-miR-26a-5p mature miRNA

- Thanks for the clarification provided. Nevertheless, I would note that measurements of the host transcript can be difficult to interpret. The processing of the hairpin by Drosha results in rapid decay of the reaming of the non-hairpin part, usually yielding very low expression levels. The mature levels of miR-26a-5p could be more accurate.

(3) Similarly, the miRNA-seq expression from isomirdb suggests also that expression of miR-26b-5p is indeed 50-fold higher than miR-26b-3p in liver and blood. This difference in abundance of the two strands are usually regarded as one of them being the guide strand (in this case the 5p) and the other being the passenger (in this case the 3p). In some cases, passenger strands can be a byproduct of miRNA biogenesis, thus the rescue experiments using LNPs with both strands on equimolar amounts would not reflect the physiological abundance miR-26b-3p. The non-physiological over abundance of miR-26b-3p would constitute a source of undesired off-targets.

- I agree with the authors that the functional data doesn't show evidence of undesired off-targets. Nevertheless, I would consider that for future studies. miRNA-phenotypes can be subtle in normal conditions and become more obvious on stressed conditions, the same might apply to off-target effects.

(4) It would also be valuable to check the miRNA levels on the liver upon LNP treatment, or at least the signatures of miR-26b-3p and miR-26b-5p activity using RNA-seq on the RNA samples already collected.

- Thanks for providing the miRNA quantification on the revised version of the manuscript.

(5) Some of the phenotypes described, such as the increase in cholesterol, overlap with the previous publication van der Vorst et al. BMC Genom Data (2021), despite in this case the authors are doing their model in Apoe knock-out and Western-type diet. I would encourage the authors to investigate more or discuss why the initial phenotypes don't become more obvious despite the stressors added in the current manuscript.

- Thanks for the clarification provided on your revised version of the manuscript.

(6) The authors have focused part of their analysis on a few gene markers that show relatively modest changes. Deeper characterization using RNA-seq might reveal other genes that are more profoundly impacted by miR-26 depletion. It would strengthen the conclusions proposed if the authors validated that changes on mRNA abundance (Sra, Cd36) do impact the protein abundance. These relatively small changes or trends in mRNA expression, might not translate into changes in protein abundance.

- Thanks for addressing this concern raised by R1 and R2.

(7) In figures 5 and 7, the authors run a phosphorylation array (STK) to analyze the changes in the activity of the kinome. It seems that a relatively big number of signaling pathways are being altered, I think that should be strengthened by further validations by Western blot on the collected tissue samples. For quite a few of the kinases there might be antibodies that recognise phosphorylation. The two figures lack a mechanistic connection to the rest of the manuscript.

- I appreciate the clarification provided by the authors regarding the difference between the activity assay and a Western blot for phosphorylated proteins. Is there any orthogonal technique to validate the PamGene activity assay available?

Comments on revised version:

The authors have addressed most of the changes suggested by R1 and R2.

---

## [Author Response]

The following is the authors’ response to the original reviews.

**Public Reviews:**

**Reviewer #1 (Public Review):**
Based on previous publications suggesting a potential role for miR-26b in the pathogenesis of metabolic dysfunction-associated steatohepatitis (MASH), the researchers aim to clarify its function in hepatic health and explore the therapeutical potential of lipid nanoparticles (LNPs) to treat this condition. First, they employed both whole-body and myeloid cell-specific miR-26b KO mice and observed elevated hepatic steatosis features in these mice compared to WT controls when subjected to WTD. Moreover, livers from whole-body miR-26b KO mice also displayed increased levels of inflammation and fibrosis markers. Kinase activity profiling analyses revealed distinct alterations, particularly in kinases associated with inflammatory pathways, in these samples. Treatment with LNPs containing miR-26b mimics restored lipid metabolism and kinase activity in these animals. Finally, similar anti-inflammatory effects were observed in the livers of individuals with cirrhosis, whereas elevated miR-26b levels were found in the plasma of these patients in comparison with healthy control. Overall, the authors conclude that miR-26b plays a protective role in MASH and that its delivery via LNPs efficiently mitigates MASH development.The study has some strengths, most notably, its employ of a combination of animal models, analyses of potential underlying mechanisms, as well as innovative treatment delivery methods with significant promise. However, it also presents numerous weaknesses that leave the research work somewhat incomplete. The precise role of miR-26b in a human context remains elusive, hindering direct translation to clinical practice. Additionally, the evaluation of the kinase activity, although innovative, does not provide a clear molecular mechanisms-based explanation behind the protective role of this miRNA.Therefore, to fortify the solidity of their conclusions, these concerns require careful attention and resolution. Once these issues are comprehensively addressed, the study stands to make a significant impact on the field.

We would like the reviewer for his/her careful evaluation of our manuscript and appreciate his/her appraisal for the strengths of our study. Regarding the weaknesses, we have addressed these as good as possible during the revision of our manuscript.

We can already state that miR-26b has clear anti-inflammatory effects on human liver slices, which is in line with our results demonstrating that miR-26b plays a protective role in MASH development in mice. The notion that patients with liver cirrhosis have increasing plasma levels of miR-26b, seems contradictory at first glance. However, we believe that this increased miR-26b expression is a compensatory mechanism to counteract the MASH/cirrhotic effects. However, the exact source of this miR-26b remains to be elucidated in future studies.

The performed kinase activity analysis revealed that miR-26b affects kinases that particularly play an important role in inflammation and angiogenesis. Strikingly and supporting these data, these effects could be inverted again by LNP treatment. Combined, these results already provide strong mechanistic insights on molecular and intracellular signalling level. Although the exact target of miR-26b remains elusive and its identification is probably beyond the scope of the current manuscript due to its complexity, we believe that the kinase activity results already provide a solid mechanistic basis.

**Reviewer #1 (Recommendations For The Authors):**
A list of recommendations for the authors is presented below:(1) The title should emphasize that the majority of experiments were conducted in mice to accurately reflect the scope of the study.

As suggested we have updated our title to include the statement that we primarily used a murine model:

“MicroRNA-26b protects against MASH development in mice and can be efficiently targeted with lipid nanoparticles.”

(2) It would be useful to know more about miR-26b function, including its target genes, tissue-specific expression, and tissue vs. circulating levels. Is it expected that the two strains of the miRNA (i.e., -3p and -5p) act this similarly? Also, miR-26b expression in the liver of individuals with cirrhosis should be determined.

The function of miR-26b is still rather elusive, making functional studies using this miR very interesting. In a previous study, describing our used mouse model (Van der Vorst et al. BMC Genom Data, 2021) we have eluded several functions of miR-26b that are already investigated. This was particularly already described in carcinogenesis and the neurological field.

Target gene wise, there are already several targets described in miRbase. However, for our experiments we feel that determination of the specific target genes is beyond the scope of the current manuscript and rather a focus of follow-up projects.

Regarding the expression of miR-26b, the liver and blood have rather high and similar expressions of both miR-26b-3p and miR-26b-5p as shown in Author response image 1.

Expression of miR-26b-3p and -5p. Expression of miR-26b-3p (left) and miR-26b-5p (right), generated by using the miRNATissueAtlas 2025 (Rishik et al. Nucleic Acids Research, 2024).

Unfortunately, due to restrictions in tissue availability and the lack of stored RNA samples, we are unable to measure miR-26b expression in the human livers. However, based on the potency of the miR-26b mimic loaded LNPs in the mice (Revised Supplemental Figure 2A-B), we are confident that these LNPs also resulted in a overexpression of miR-26b in the human livers.

(3) Please explain the rationale behind primarily using whole-body miR-26b KO mice rather than the myeloid cell-specific KO model for the studies.

The main goal of our study is the elucidation of the general role of miR-26b in MASH formation. Therefore, we decided to primarily focus on the whole-body KO model. While we used the myeloid cell-specific KO model to highlight that myeloid cells play an important role in the observed phenotypes, we believe the whole-body KO model is more appropriate as main focus, particularly also in light of the used LNP targeting that also provides a whole-body approach. Furthermore, this focus on the whole-body model also reflects a more therapeutically relevant approach.

(4) The authors claim that treatment with LNPs containing miR-26b "replenish the miR-26b level in the whole-body deficient mouse" but the results of this observation are not presented.

This is indeed a valid point that we have now addressed. We have measured the mir26b-3p and mir26b-5p expression levels in livers from mice after 4-week WTD with simultaneous injection with either empty LNPs as vehicle control (eLNP) or LNPs containing miR-26b mimics (mLNP) every 3 days. As shown in Revised Supplemental Figure 2A-B, mLNP treatment clearly results in an overexpression of the mir26b in the livers of these mice. We have rephrased the text accordingly by stating that mLNP results in an “overexpression” rather than “replenishment”.

(5) The number of 3 human donors for the precision-cut liver slices is clearly insufficient and clinical parameters need to be shown. Additionally, inconsistencies in individual values in Figures 8B-E need clarification.

Unfortunately, due to restrictions in tissue availability, we are unable to increase our n-number for these experiments. Clinical parameters are not available, but the liver slices were from healthy tissue.

We have performed these experiments in duplicates for each individual donor. We have now specified this also in the figure legend to explain the individual values in the graphs:

“…(3 individual donors, cultured in duplicates).”

(6) Figure 2D: Please include representative images.

As suggested we have included representative images in our revised manuscript.

(7) Address discrepancies in the findings across different experimental settings. For example, the expression levels of the lipid metabolism-related genes are not significantly modulated in whole-body miR-26b KO mice (except for Sra), but they are in the myeloid cell-specific model (but not Sra), and none of them are restored after LNPs injections.

Although *Cd36* is not significantly increased in the whole-body miR-26b KO mice, there is a clear tendency towards increased expression, which is now also validated on protein level (Revised Figure 1K-L). In the myeloid cell-specific model we see a similar tendency, although the gene expression difference of *Sra* is not significantly changed. This could be due to the difference in the model, since only myeloid cells are affected, suggesting that the effects on *Sra* are to a large extend driven by non-myeloid cells. This would also fit to the tendency to decreased *Sra* expression in the mimic-LNP treated mice. Due to the larger variation, this difference did not reach significance, which is rather a statistical issue due to relatively small n-numbers. At this moment, we cannot exclude that these receptors are differentially regulated by different cell-types. For this, future studies are needed focussing on cell-specific targeting of miR-26b in somatic cells, like hepatocytes.

(8) Figure 4A the images are not representative of the quantification.

We have selected another representative image that is exactly reflecting the average Sirius red positive area, to reflect the quantification appropriately.

(9) Figures 5 and 7: Are there not significantly decreased/increased kinases? A deeper analysis of these kinase alterations is necessary to understand how miR-26b exerts its role. A comparison analysis of these two datasets might clarify this regard.

We indeed very often see in these kinome analysis that the general tendency of kinase activity is unidirectional. We believe that this is caused by the highly interconnected nature of kinases. Activation of one signalling cascade will also results in the activation of many other cascades. However, it is interesting to see which pathways are affected in our study and we find it particularly interesting to see that the tendencies is exactly opposite between both comparisons as KO vs. WT shows increase kinase activities, while KO-LNP vs. KO shows a decrease again. Further showing that the method is reflecting a true biological effect that is mediated by miR26b.

(10) Determinations of the effect of LNPs containing miR-26b in the KO mice are limited to only a few observations (that are not only significant). More extensive findings are needed to conclusively demonstrate the effectiveness of this treatment method. Similar to the experiments with human liver samples (Figures 8A-E).

We have now elaborated our observations in the mouse model using LNPs by also analysing the effects on inflammation and fibrosis. However, it seems that the treatment time was not long enough to see pronounced changes on these later stages of disease development. Interestingly, the expression of *Tgfb* was significantly reduced, suggesting at least that the LNPs on genetic levels have an effect already on fibrotic processes. Thereby, it can be suggested that longer mLNP treatment may result in more effects on protein level as well, which remains to be determined in future studies.

Unfortunately, due to restrictions in tissue availability, we are unable to increase our n-number or read-outs for these experiments at this moment.

(11) In Figures 8F-H, the observed increase in circulating miR-26b levels in the plasma of cirrhotic individuals seems contradictory to its proposed protective role. This discrepancy requires clarification.

In the revised discussion (second to last paragraph), we have now elaborated more on the findings in the plasma of cirrhotic individuals in comparison to our murine *in-vivo* results, to highlight and discuss this discrepancy.

(12) Figures 8F-H legend mentions using 8-11 patients per group, but the methods section lacks corresponding information about these individuals.

These patients, together with inclusion/exclusion criteria and definition of cirrhosis are described in the method section 2.14.

(13) Figure 8G has 7 data points in the cirrhosis group, instead of 8. Any data exclusion should be justified in the methods section.

As defined in method section 2.15, we have identified outliers using the ROUT = 1 method, which is the reason why Figure 8G only has 7 data points instead of 8.

**Reviewer #2 (Public Review):**
Summary:This manuscript by Peters, Rakateli, et al. aims to characterize the contribution of miR-26b in a mouse model of metabolic dysfunction-associated steatohepatitis (MASH) generated by a Western-type diet on the background of Apoe knock-out. In addition, the authors provide a rescue of the miR-26b using lipid nanoparticles (LNPs), with potential therapeutic implications. In addition, the authors provide useful insights into the role of macrophages and some validation of the effect of miR-26b LNPs on human liver samples.Strengths:The authors provide a well-designed mouse model, that aims to characterize the role of miR-26b in a mouse model of metabolic dysfunction-associated steatohepatitis (MASH) generated by a Western-type diet on the background of Apoe knock-out. The rescue of the phenotypes associated with the model used using miR-26b using lipid nanoparticles (LNPs) provides an interesting avenue to novel potential therapeutic avenues.Weaknesses:Although the authors provide a new and interesting avenue to understand the role of miR-26b in MASH, the study needs some additional validations and mechanistic insights in order to strengthen the author's conclusions.(1) Analysis of the expression of miRNAs based on miRNA-seq of human samples (see https://ccb-compute.cs.uni-saarland.de/isomirdb/mirnas) suggests that miR-26b-5p is highly abundant both on liver and blood. It seems hard to reconcile that despite miRNA abundance being similar in both tissues, the physiological effects claimed by the authors in Figure 2 come exclusively from the myeloid (macrophages).

We agree with the reviewer that the effects observed in the whole-body KO model are most likely a combination of cellular effects, particularly since miR-26b is also highly expressed in the liver. However, with the LysM-model we merely want to demonstrate that the myeloid cells at least play an important, though not exclusive, role in the phenotype. In the discussion, we also further elaborate on the fact that the observed changes in the liver can me mediated by hepatic changes.

To stress this, we have adjusted the conclusion of Figure 2:

“Interestingly, mice that have a myeloid-specific lack of miR-26b also show increased hepatic cholesterol levels and lipid accumulation demonstrated by Oil-red-O staining, coinciding with an increased hepatic Cd36 expression (Figure 2), demonstrating that myeloid miR-26b plays a major, but not exclusive, role in the observed steatosis.”

(2) Similarly, the miRNA-seq expression from isomirdb suggests also that expression of miR-26a-5p is indeed 4-fold higher than miR-26b-5p both in the liver and blood. Since both miRNAs share the same seed sequence, and most of the supplemental regions (only 2 nt difference), their endogenous targets must be highly overlapped. It would be interesting to know whether deletion of miR-26b is somehow compensated by increased expression of miR-26a-5p loci. That would suggest that the model is rather a depletion of miR-26.UUCAAGUAAUUCAGGAUAGGU mmu-miR-26b-5p mature miRNAUUCAAGUAAUCCAGGAUAGGCU mmu-miR-26a-5p mature miRNA

This is a very valid point raised by the reviewer, which we actually already explored in a previous study, describing our used mouse model (Van der Vorst et al. BMC Genom Data, 2021). In this manuscript, we could show that miR-26a is not affected by the deficiency of miR-26b (Figure 1G in: Van der Vorst et al. BMC Genom Data, 2021).

(3) Similarly, the miRNA-seq expression from isomirdb suggests also that expression of miR-26b-5p is indeed 50-fold higher than miR-26b-3p in the liver and blood. This difference in abundance of the two strands is usually regarded as one of them being the guide strand (in this case the 5p) and the other being the passenger (in this case the 3p). In some cases, passenger strands can be a byproduct of miRNA biogenesis, thus the rescue experiments using LNPs with both strands in equimolar amounts would not reflect the physiological abundance miR-26b-3p. The non-physiological overabundance of miR-26b-3p would constitute a source of undesired off-targets.

We agree with the reviewer on this aspect and this is something we had to consider while generating the mimic LNPs. However, we believe that we do not observe and undesired off-target effects, as the effects of the mimic LNPs at least on functional outcomes are relatively mild and only restricted to the expected effects on lipids. Furthermore, the effects on the kinase profile due to the mimic LNP treatment are in line with our expectations. Combined these results suggest at least that potential off-target effects are minor.

(4) It would also be valuable to check the miRNA levels on the liver upon LNP treatment, or at least the signatures of miR-26b-3p and miR-26b-5p activity using RNA-seq on the RNA samples already collected.

This is indeed a valid point that we have now addressed. We have measured the mir26b-3p and mir26b-5p expression levels in livers from mice after 4-week WTD with simultaneous injection with either empty LNPs as vehicle control (eLNP) or LNPs containing miR-26b mimics (mLNP) every 3 days. As shown in Supplemental Figure 2A-B, mLNP treatment clearly results in an overexpression of the mir26b in the livers of these mice. We have rephrased the text accordingly by stating that mLNP results in an “overexpression” rather than “replenishment”.

(5) Some of the phenotypes described, such as the increase in cholesterol, overlap with the previous publication by van der Vorst et al. BMC Genom Data (2021), despite in this case the authors are doing their model in Apoe knock-out and Western-type diet. I would encourage the authors to investigate more or discuss why the initial phenotypes don't become more obvious despite the stressors added in the current manuscript.

In our previous publication (BMC Genom Data; 2021), we actually did not see any changes in circulating lipid levels. However, in that study we did not evaluate the livers of the mice, so we do not have any information about the hepatic lipid levels.

As mentioned by the reviewer, we believe that we see much more pronounced phenotypes in the current model because we use the combined stressor of *Apoe-/-* and Western-type diet, which cannot be compared to the wildtype and chow-fed mice used in the BMC Genom Data manuscript.

(6) The authors have focused part of their analysis on a few gene makers that show relatively modest changes. Deeper characterization using RNA-seq might reveal other genes that are more profoundly impacted by miR-26 depletion. It would strengthen the conclusions proposed if the authors validated that changes in mRNA abundance (Sra, Cd36) do impact the protein abundance. These relatively small changes or trends in mRNA expression, might not translate into changes in protein abundance.

As suggested by the reviewer we have now also confirmed that the protein expression of CD36 and SRA is significantly increased upon miR-26b depletion, visualized as Figure 1K-L in the revised manuscript. Unfortunately, we do not have enough material left to perform similar analysis for the LysM-model or the LNP-model, although based on the whole-body effects we are confident that at least for CD36/SRA in this case the gene expression matches effects observed on protein level.

(7) In Figures 5 and 7, the authors run a phosphorylation array (STK) to analyze the changes in the activity of the kinome. It seems that a relatively large number of signaling pathways are being altered, I think that should be strengthened by further validations by Western blot on the collected tissue samples. For quite a few of the kinases, there might be antibodies that recognise phosphorylation. The two figures lack a mechanistic connection to the rest of the manuscript.On this point we respectfully have to disagree with the reviewer. We have used a kinase activity profiling approach (PamGene) to analyse the real-time activity of kinases in our lysates. This approach is different than the classical Western blot approach in which only the presence or absence of a specific phosphorylation is detected. Thereby, Western blot analysis does not analyse phosphorylation in real-time, but rather determines whether there has been phosphorylation in the past. Our approach actually determines the real-time, current activity of the kinases, which we believe is a different and perhaps even more reliable read-out measurement. Therefore, validation by Western blot would not strengthen these observations.

We have particularly tried to connect these observations to the rest of the manuscript by highlighting the observed signalling cascades that are affected, highlighting a role in inflammation and angiogenesis, thereby providing some mechanistic insights.

**Reviewer #2 (Recommendations For The Authors):**
I would encourage the authors to follow-up on some of the more miRNA focused comments made above, which would strengthen the mechanistic part of the work presented.I suggest the authors tone down some of some of the claims made (eg. "clearly increased expression", "exacerbated hepatic fibrosis"), given that some of it might need further validation.

Wherever needed we have tuned down the tone of some claims, although we believe that most claims are already written carefully enough and in line with the observed results.

Some of the panels that are supposed to have the same amount of animals have variable N, despite they come from the same exact number of RNA samples or tissue lysates (eg. 1G and 1H, vs 1I and 1J).

This is indeed correct and caused by the fact that some analysis resulted in statistical outliers as identified using the ROUT = 1 method, as also specified in section 2.15 of the method section.

It would be nice to have representative images of oil-red-o in all the figures where it is quantified (or at least in the supplementary figures).

As suggested by the reviewer, we have now included representative images for the LysM-model (Revised Figure 2D) and the LNP-model (Revised Figure 6D) as well.